# Neural evidence accumulation persists after choice to inform metacognitive judgments

Peter R Murphy[1,2]*, Ian H Robertson[1], Siobhán Harty[1], Redmond G O'Connell[1]

[1]Trinity College Institute of Neuroscience, School of Psychology, Trinity College Dublin, Dublin, Ireland; [2]Institute of Psychology, Leiden Institute for Brain and Cognition, Leiden University, Leiden, The Netherlands

**Abstract** The ability to revise one's certainty or confidence in a preceding choice is a critical feature of adaptive decision-making but the neural mechanisms underpinning this metacognitive process have yet to be characterized. In the present study, we demonstrate that the same build-to-threshold decision variable signal that triggers an initial choice continues to evolve after commitment, and determines the timing and accuracy of self-initiated error detection reports by selectively representing accumulated evidence that the preceding choice was incorrect. We also show that a peri-choice signal generated in medial frontal cortex provides a source of input to this post-decision accumulation process, indicating that metacognitive judgments are not solely based on the accumulation of feedforward sensory evidence. These findings impart novel insights into the generative mechanisms of metacognition.

*For correspondence: murphyp7@tcd.ie

Competing interests: The authors declare that no competing interests exist.

## Introduction

The ability to detect errors is an essential feature of adaptive behavior, providing the basis for adjusting or countermanding ongoing actions and optimizing future decision-making (*David et al., 2012*; *Fernandez-Duque et al., 2000*; *Fleming et al., 2012*). Establishing the neurocomputational principles underpinning this key metacognitive function is therefore a major imperative. Categorical choices are thought to be made by integrating evidence over time into a decision variable that triggers action upon reaching a criterion (*Gold and Shadlen, 2007*; *Kelly and O'Connell, 2014*; *Shadlen and Kiani, 2013*; *Smith and Ratcliff, 2004*). Most theoretical models of metacognition propose that the same decision variable makes a key contribution to internal representations of choice accuracy (*de Martino et al., 2013*; *Heath, 1984*; *Kiani et al., 2014*; *Kiani and Shadlen, 2009*; *Link, 2003*; *Moran et al., 2015*; *Pleskac and Busemeyer, 2010*; *Ratcliff and Starns, 2013*; *Yu et al., 2015*). However, there is considerable uncertainty regarding the precise nature of this contribution.

Initial efforts to model metacognitive performance centered on the proposition that our confidence in a choice is based on a read-out of the level the decision variable has reached at the time of choice commitment (*Heath, 1984*; *Kiani et al., 2014*; *Kiani and Shadlen, 2009*; *Link, 2003*). However, the specification that the decision variable reaches a fixed threshold prior to commitment means that these models cannot account for the fact that human participants can retrospectively categorize certain choices as erroneous even in the absence of external feedback (*Rabbitt and Vyas, 1981*; *Rabbitt, 1966*; *Yeung et al., 2004*). To take account of this kind of observation, alternative theoretical models have been proposed in which metacognitive judgments can exploit additional evidence that is accumulated after first-order commitment (*Moran et al., 2015*; *Pleskac and Busemeyer, 2010*; *Yu et al., 2015*). However, a definitive neurophysiological demonstration of post-

**eLife digest** Reflecting on our previous choices and accurately representing our confidence in their accuracy allows us to detect, correct and learn from our errors. Yet, it remains poorly understood how such "metacognition", or *thoughts about thoughts*, occurs in the human brain. In particular, a long-standing debate in this area of research concerns whether metacognitive processes in the brain occur at the same time as those that determine the actual choice, or whether they develop after the choice has been made and rely on different information.

Now, Murphy et al. have recorded brain activity in human volunteers who were carrying out a simple task in order to explore metacognition. In short, the volunteers looked at colored words and decided if each word matched its color (e.g., is the word 'RED' also written in a red font?). At the same time, the volunteers chose whether or not to press a button depending on the specific color/word combination shown, and most importantly reported whenever they noticed that they made an error in the task.

This approach allowed Murphy et al. to chart the development of choices and detection of errors as they occurred in the volunteers' brains. This revealed that the metacognitive judgement about each choice relied on information that was gathered after the point the initial choice was made. Further analysis then suggested that this process relies, at least in part, on a signal generated in a region at the front of the brain.

Together, these findings suggest that metacognitive decisions rely on processes that are similar to those behind other decisions, but with a few important differences. Namely, the metacognitive process plays out at a different point in time, and likely incorporates distinct sources of information. Further work should aim to clarify the nature of these sources of information and describe their specific contributions to the process.

decisional evidence accumulation has yet to be definitively provided in humans or other animals. Moreover, it is unclear whether such a process would take the form of a simple continuation of first-order evidence gathering (*Moran et al., 2015*; *Pleskac and Busemeyer, 2010*; *Resulaj et al., 2009*; *Yu et al., 2015*) or might also be crucially dependent on higher-order representations of error likelihood (*Yeung et al., 2004*).

While the last two decades have seen intensive research on the neural signatures of decision formation in the non-human primate (*Gold and Shadlen, 2007*; *Shadlen and Kiani, 2013*), these questions have proven difficult to address because the pre-motor neurons that have been the focus of much of this work (e.g. in area LIP) fall silent upon initiation of the decision-reporting action, thus precluding measurement of post-decision activity and verification of its potential influence on metacognition. Although post-commitment neural signatures have been described in humans (*Falkenstein et al., 1990*; *Gehring et al., 1993*) and other animals (*Ito et al., 2003*; *Narayanan et al., 2013*; *Pardo-Vazquez et al., 2008*) which show clear sensitivity to choice accuracy and, in some cases, to the quality of metacognitive judgments (*Boldt and Yeung, 2015*; *Nieuwenhuis et al., 2001*; *O'Connell et al., 2007*; *Steinhauser and Yeung, 2010*), these signals have yet to be conclusively identified with post-decisional evidence integration. In the present study, we exploited the uniquely supramodal nature of a recently characterized build-to-threshold decision variable signal in the human brain (*Kelly and O'Connell, 2013*; *O'Connell et al., 2012*; *Twomey et al., 2015*) to investigate the influence of post-decisional evidence accumulation on the timing and accuracy of explicit error detection. Through a combination of electrophysiological data analysis and computational modeling, we demonstrate that neural evidence accumulation does persist after commitment to the first-order decision and that its rate determines the probability and timing of error detection. Additionally, we show that the rate of post-decision accumulation is not solely the product of feedforward sensory inputs but is influenced by the output of medial frontal structures implicated in performance monitoring and executive control.

# Results

## Task behavior

We analyzed 64-channel electroencephalographic (EEG) data [originally collected as part of *Murphy et al. (2012)*; see *Materials and methods*], from 28 human subjects performing a Go/No-Go response inhibition task (*Hester et al., 2005*; *Figure 1a*). Subjects viewed a serial sequence of color words, each presented for 0.4 s, with the congruency between font color and semantic content varied across trials. The primary task was to execute a right-handed button press as quickly as possible when the semantic content of the word and its font color were incongruent (Go trial), and to withhold this response when either the word presented on the current trial was the same as that presented on the previous trial ('repeat' No-Go) or when the meaning of the word and its font color matched ('color' No-Go). Performance on paradigms of this nature can be readily understood in

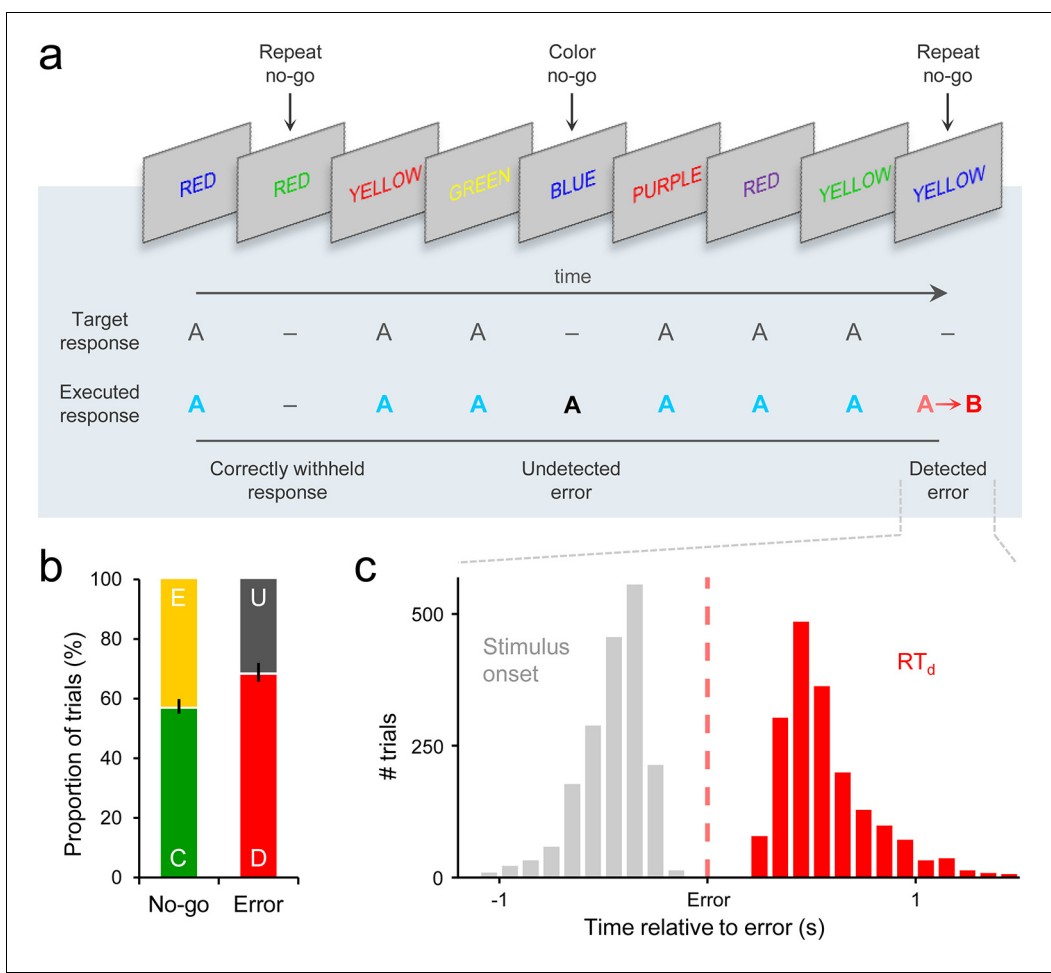

**Figure 1.** Go/No-Go task and associated behavior. (**a**) Subjects' primary task was to make a speeded manual response ('A') to all incongruent color/word stimuli and to withhold from responding to congruent stimuli or when the same word was presented on consecutive trials. Following any commission errors, they were instructed to signal error detection as quickly as possible by pressing a secondary response button ('B'). (**b**) Distribution of trial types averaged across subjects. C = correct withhold, E = error, D = detected error, U = undetected error. Error bars = s.e.m. (**c**) Histograms representing stimulus onset and detection response time (RT$_d$) distributions on detected error trials, aligned to error commission and pooled across all subjects.

The following figure supplement is available for figure 1:

**Figure supplement 1.** 'Repeat' and 'color' No-Go stimuli did not differentially affect primary RT, RT$_d$ or decision signal morphology on detected error trials.

terms of a race-to-threshold between two competing accumulation processes representing the evidence in favor of a 'Go' or 'Don't Go' decision (*Gomez et al., 2007*; *Logan and Cowan, 1984*).

If subjects failed to withhold the pre-potent response to either type of No-Go stimulus, they were instructed to signal detection of this error as quickly as possible by pressing a secondary response button. The timing of this error detection report was measured relative to the initial erroneous action (detection response time; $RT_d$). There were no significant differences in primary RT, $RT_d$ or electrophysiological signal morphology between repeat and color No-Go trials (*Figure 1—figure supplement 1*); therefore, we collapsed across both No-Go trial-types in all analyses.

Subjects successfully withheld from responding on an average of 56.6% (±13.0) of No-Go trials, and 68.0% (±16.7) of erroneous presses were followed by an error detection report (*Figure 1b*). Primary RT and $RT_d$ were not correlated across detected error trials (mean within-subject $\beta = -0.05$, s. e.m. = 0.04; $t_{27} = -1.4$, $p = 0.2$). The median primary RTs for detected errors (457 ± 21 ms) were faster than correct Go RTs (511 ± 22 ms; $t_{27} = 5.5$, $p < 1 \times 10^{-4}$), whereas undetected errors were significantly slower than Go RTs (543 ± 28 ms; $t_{27} = 2.9$, $p = 0.007$; detected vs. undetected errors: $t_{27} = 5.8$, $p < 1 \times 10^{-4}$).

## A centro-parietal signature of first- and second-order evidence accumulation

A series of recent studies of perceptual decision making have established that a decision variable signal can be isolated in the human event-related potential (ERP) over centro-parietal scalp sites (*Kelly and O'Connell, 2013*; *O'Connell et al., 2012*; *Twomey et al., 2015*). This signal exhibits the same decision-predictive dynamics that have been reported in single-unit recordings from a variety of brain areas during perceptual decision formation (*Gold and Shadlen, 2007*; *Shadlen and Kiani, 2013*), including an evidence-dependent rate of rise and a threshold-crossing relationship with reaction time. Another important feature of this signal is that it represents the evolving decision in a domain-general fashion that is independent of motor requirements and indeed traces the decision even when no overt decision-reporting action is required (*O'Connell et al., 2012*). Here, we observed that the same centro-parietal positivity (CPP) was elicited by Go and No-Go stimuli (*Figure 2—figure supplement 1*). In what follows, we demonstrate that this signal encodes consecutive build-to-threshold processes that determine both first-order choices and second-order error detection decisions on our task. Readers familiar with the human ERP literature will note that these two processing stages incorporate stimulus- and response-evoked activity usually attributed to the 'P300' and 'Pe' components, respectively (see *Discussion*). We persist here with the label 'CPP' because it is less prescriptive about signal latency, eliciting conditions or measurement technique.

To probe the dynamics of the CPP and its relationship to the timing of the first-order decision process, we split each subject's Go-trial RT distribution into equal-sized fast, medium and slow bins and plotted the average waveforms aligned to action execution for each bin. Consistent with previous observations (*Kelly and O'Connell, 2013*; *O'Connell et al., 2012*; *Twomey et al., 2015*), the CPP exhibited a gradual build-up with a rate that was inversely proportional to RT, and reached a stereotyped amplitude at the time of first-order choice commitment (*Figure 2a*). Thus, first-order performance on the Go/No-Go task was reliant on the same fundamental neural dynamics as have been reported for conventional perceptual decision-making paradigms (*Kelly and O'Connell, 2013*).

Despite the fact that the median $RT_d$ was executed 560 ms after stimulus offset, precluding the continued integration of feedforward sensory evidence in the period preceding error detection, the CPP exhibited persistent post-commitment build-up on a subset of trials, continuing its positive trajectory prior to an error detection report but gradually diminishing in amplitude following Go decisions and undetected errors (*Figure 2b*; see *Figure 3—figure supplement 1* for explicit comparison of signals on Go and undetected error trials). Receiver operating characteristic (ROC) curve analysis (see *Materials and methods*) conducted in discrete temporal windows along the entire signal time course revealed that second-order performance could be reliably classified as early as 120 ms prior to error commission (*Figure 2c*).

We next sought to characterize the relationship between the CPP and the timing of error detection by splitting each subject's $RT_d$ distribution into three equal-sized bins and plotting the bin-averaged waveforms aligned to both error commission and the error detection report. Mirroring our observations for the first-order responses, the build-up rate of the second-order CPP was steeper on

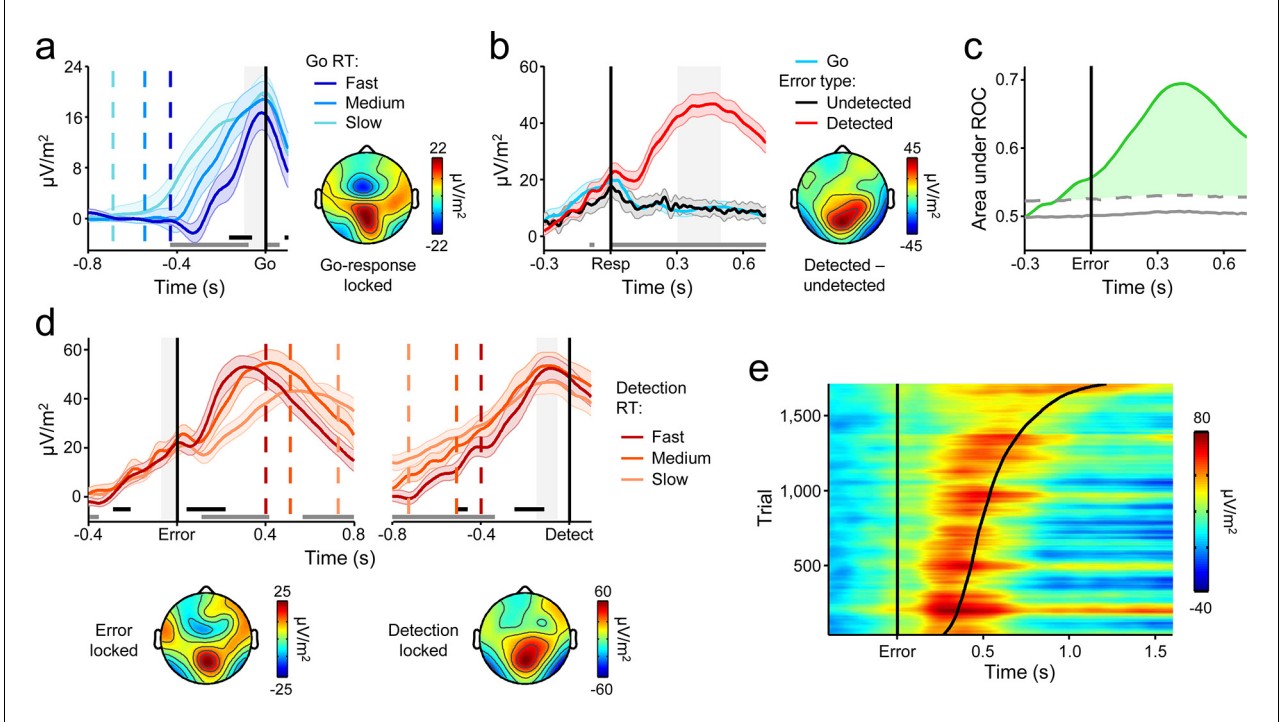

**Figure 2.** A centro-parietal decision signal for first- and second-order decision-making. (**a**) Go-trial CPP waveforms aligned to primary task response and sorted by RT into three equal-sized bins, and associated scalp topography. (**b**) CPPs again aligned to primary response, but separately for Go trials and detected and undetected errors; gray running marker indicates significant detection effect ($p < 0.05$, paired $t$-test for detected vs. undetected). Topography illustrates scalp distribution of error detection effect. (**c**) Time-course of detection-predictive activity estimated as the area under the ROC curve. Permutation mean and significance threshold (1.96 s.d.) are marked as solid and dashed gray lines, respectively. (**d**) Detected-error CPP, aligned to primary task response and subsequent error detection report; waveforms were sorted and binned by $RT_d$. (**e**) Single-trial surface plot showing temporal relationship between the CPP and $RT_d$ (curved black line); waveforms were pooled across subjects, sorted by $RT_d$ and smoothed over bins of 50 trials with Gaussian-weighted moving average. Vertical dashed lines in a and d represent median RTs. Gray markers at bottom of these plots indicate time points when linear regression of RT on signal amplitude reached significance ($p < 0.05$); black markers indicate center of 150 ms time windows in which regression of RT on signal slope reached significance ($p < 0.05$; one-tailed predicting steeper slope for faster RTs). Shaded gray areas show latencies of all associated scalp topographies. All traces were baselined to pre-stimulus period. Shaded error bars = s.e.m.

The following figure supplements are available for figure 2:

**Figure supplement 1.** The CPP aligned to stimulus onset, separately for Go and correctly withheld No-Go trials.

**Figure supplement 2.** Early stimulus-evoked deflections affect first-order CPP amplitude on fast Go trials.

trials characterized by faster error detection, and it again reached a fixed amplitude immediately prior to the error detection report (*Figure 2d,e*).

Taken together, these findings indicate that subjects engaged in evidence accumulation even after the primary task response had been executed and leveraged the new information gained by this process to make judgments about the accuracy of their preceding choices. However, CPP dynamics during the post-commitment interval were qualitatively distinct from those observed during the first-order decision process. While the positive build-up of the first-order CPP was invariant to trial-type, tracing the emerging decision irrespective of whether the evidence favored a Go or No-Go choice (*Figure 2—figure supplement 1*; see also *Kelly and O'Connell, 2013*), the second-order CPP only increased after detected errors and not, on average, after correct Go responses or undetected errors. Similarly, in a recent study that examined the post-commitment 'Pe' signal preceding graded confidence judgments (without interrogating this signal for accumulation-to-bound characteristics), a monotonic relationship between signal amplitude and confidence was observed whereby amplitude was greatest when subjects indicated very *low* confidence in the primary choice

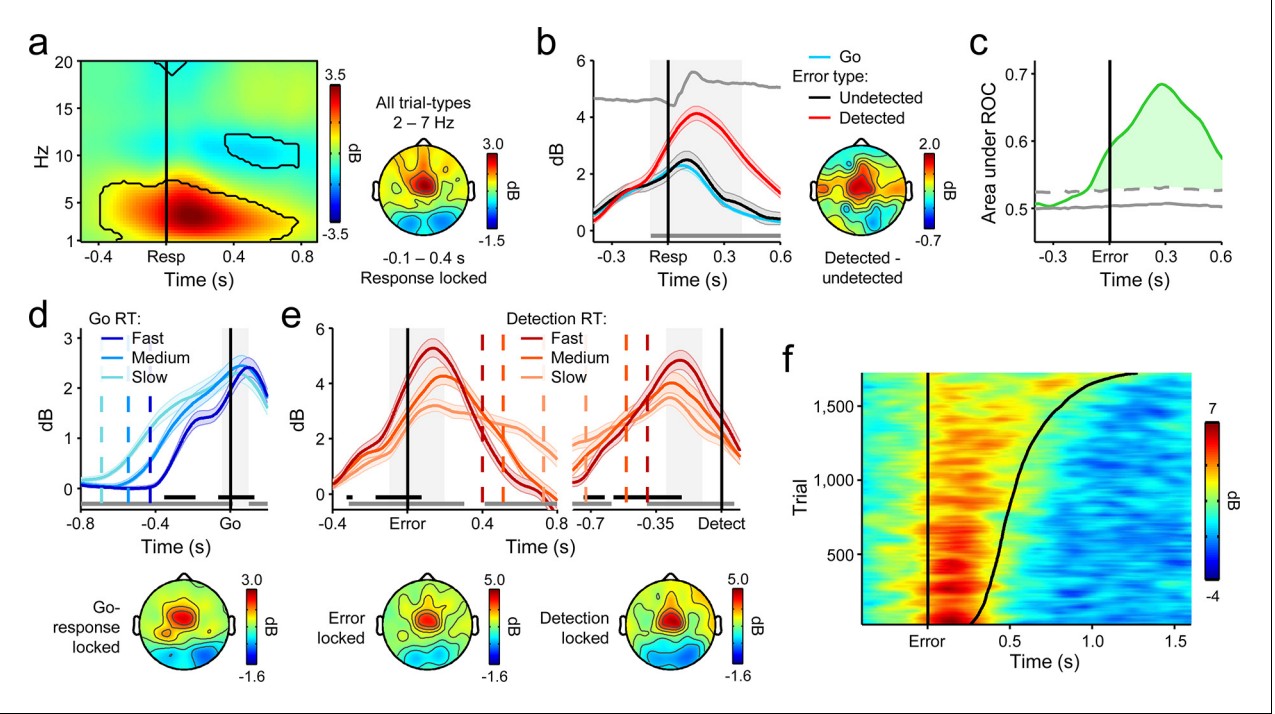

**Figure 3.** Fronto-central θ-band (2–7 Hz) oscillatory power predicts the accuracy and timing of error detection reports. (**a**) Time-frequency plot of fronto-central power, aligned to the primary task response and averaged across detected error, undetected error and RT-matched go trials; black lines enclose regions of significant power change relative to a pre-stimulus baseline ($p < 0.01$, paired $t$-test). Scalp topography shows θ power averaged over all trial types. (**b**) Response-aligned FCθ waveforms separately for each trial type; gray trace is the condition-averaged ERP (arbitrarily scaled). Topography illustrates scalp distribution of error detection effect. (**c**) Time-course of FCθ detection-predictive activity quantified by the area under the ROC curve. (**d**) Go-trial FCθ waveforms and associated scalp topography, aligned to primary task response; single-trial waveforms were sorted by primary RT and divided three equal-sized bins. (**e**) Detected-error FCθ waveforms and topographies, aligned to both the primary task response (*left*) and the subsequent error detection report (*right*); waveforms were sorted and binned by $RT_d$. (**f**) Single-trial surface plot showing the temporal relationship between the FCθ power and $RT_d$ (curved black line). Conventions for b–f are the same as in *Figure 2*. All traces were again baselined to the pre-stimulus period. Plots in a were calculated via wavelet convolution; all other plots show filter-Hilbert transformed data (see *Materials and methods*).

The following figure supplements are available for figure 3:

**Figure supplement 1.** Comparison of response-aligned CPP (**a**), fronto-central ERP (**b**) and FCθ (**c**) signals on correct Go trials and undetected errors.

**Figure supplement 2.** Broadband ERPs averaged over fronto-central electrodes FCz, F1 and F2 and aligned to the primary response, separately for Go trials and detected and undetected errors.

**Figure supplement 3.** Robustness of the FCθ effects to primary RT differences and baselining regimes.

(*Boldt and Yeung, 2015*). Thus, rather than reflecting a continuation of the first-order decision process, these data suggest that error detection decisions were based on the selective accumulation of internal evidence that the previous choice was incorrect. In the following set of analyses, we investigated whether a higher-order neural signal that has known sensitivity to choice accuracy might influence the error detection process.

## Fronto-central theta oscillations and error detection behavior

A substantial literature incorporating several species and modes of neurophysiological measurement has established that the posterior medial frontal cortex (pMFC) is highly responsive to variations in performance accuracy (*Carter et al., 1998*; *Ito et al., 2003*; *Narayanan et al., 2013*; *Ridderinkhof et al., 2004*) and its activity predicts neural and behavioral adaptation following error commission (*Cavanagh et al., 2009*; *Danielmeier et al., 2011*; *Ebitz and Platt, 2015*;

*Narayanan et al., 2013*; *Sheth et al., 2012*). These characteristics are thought to reflect the pMFC's role in coordinating the activity of task-relevant regions in the presence of increasing conflict between incompatible actions (*Botvinick et al., 2001*; *Carter and van Veen, 2007*; *Cavanagh et al., 2009*; *Kerns et al., 2004*), while the conflict signal generated in this brain region has also been proposed to provide a reliable basis for subsequent error detection (*Yeung et al., 2004*). Accordingly, we investigated the influence of pMFC on error detection decisions by interrogating a prominent oscillatory signature that provides a proxy for pMFC activity, fronto-central theta power (FC$\theta$; 2–7 Hz; *Cavanagh and Frank, 2014*; *Cavanagh et al., 2011*; *Cohen, 2014*; *Cohen and Donner, 2013*; *Narayanan, et al., 2013*).

Consistent with previous reports (*Cavanagh et al., 2009*; *Cohen and Donner, 2013*; *Narayanan et al., 2013*), we observed a clear increase in FC$\theta$ power following stimulus onset that peaked after primary response execution (*Figure 3a*). The time-course of FC$\theta$ did not distinguish between correct Go responses and undetected errors (*Figure 3—figure supplement 1*) but underwent a sharp positive deflection at approximately the time of errors that were subsequently detected (*Figure 3b*). ROC analysis applied to single-trial FC$\theta$ waveforms revealed strong detection-predictive activity that, as with the CPP, achieved significant detection classification up to 120 ms before initial error commission (*Figure 3c*). This sensitivity to error detection was not apparent in the error-related negativity, a commonly investigated error-evoked ERP over fronto-central scalp (*Falkenstein et al., 1990*; *Gehring et al., 1993*; *Figure 3—figure supplement 2*), and was not due to condition-related differences in primary RT (*Figure 3—figure supplement 3a*).

We repeated the trial-binning analyses that were previously applied to the CPP, this time to explore the relationship between the FC$\theta$ signal and the timing of first- and second-order decision-making. The amplitude of the FC$\theta$ signal consistently distinguished between RT bins throughout both decision intervals (*Figure 3d,e,f*). In keeping with previous observations (*Cavanagh et al., 2011*; *Cohen and Donner, 2013*), pre-response FC$\theta$ power was consistently lower on trials with faster first-order RTs (*Figure 3d*). A reliable relationship also existed between pre-detection FC$\theta$ on detected errors and RT$_d$, but here fast error detections were preceded by *greater* FC$\theta$ power (*Figure 3e,f*). The latter observation is consistent with the notion that peri-response pMFC activity serves as a form of input to the second-order decision process and thereby influences the probability and timing of error detection (*Yeung et al., 2004*). Next, we further explored this possibility by examining the trial-by-trial prediction of RT$_d$ by different features of the FC$\theta$ and CPP signals.

## Signal interactions predict the timing of error detection

We quantitatively compared the independent contributions of the FC$\theta$ and CPP signals to variation in RT$_d$ via single-trial, within-subjects robust regressions, leveraging single-trial amplitudes, build-up rates and peak latencies of the signals as predictors in successive models (see *Materials and methods*; see *Figure 4* for measurement approach). As expected, single-trial FC$\theta$ power was strongly negatively related to RT$_d$ ($t_{27} = -7.2$, $p < 1 \times 10^{-6}$). Topographically, this effect was maximal over fronto-central scalp (*Figure 4a*). In contrast, the pre-detection amplitude of the second-order CPP did not reliably correlate with RT$_d$ ($t_{27} = -1.9$, $p = 0.08$; *Figure 4—figure supplement 1a*), consistent with the aforementioned observation of a threshold-crossing effect prior to error detection. Statistical comparison of the associated regression weights indicated that FC$\theta$ power was a better predictor of RT$_d$ than CPP amplitude ($t_{27} = 3.3$, $p = 0.003$). The opposite pattern was apparent for the build-up rates and peak latencies of both signals: CPP build-up rate ($t_{27} = -5.0$, $p < 1 \times 10^{-4}$) and latency ($t_{27} = 4.9$, $p < 1 \times 10^{-4}$, derived by permutation testing; see *Materials and methods*) robustly predicted RT$_d$ with both effects maximal over centro-parietal scalp (*Figure 4b,c*), whereas these features of the FC$\theta$ signal did not reliably account for trial-by-trial variance in RT$_d$ (build-up rate: $t_{27} = -1.9$, $p = 0.07$; latency: $t_{27} = -0.7$, $p = 0.5$; *Figure 4—figure supplement 1b,c*). Thus, although a relationship between FC$\theta$ build-up rate and the timing of error detection was clearly observed in the trial-averaged waveforms (*Figure 3e*), single-trial regressions revealed that this effect was only marginally significant when the contribution of CPP build-up rate was accounted for. Formal comparisons of the regression weights confirmed that CPP build-up rate and peak latency were superior predictors of RT$_d$ compared to the counterpart metrics derived from FC$\theta$ (build-up rate: $t_{27} = 3.0$, $p=0.006$; latency: $t_{27} = 4.3$, $p = 0.0002$).

Two additional features of our data point to an active role for FC$\theta$ in the error detection process. First, FC$\theta$ power began to reliably predict the speed of error detection approximately 350 ms before

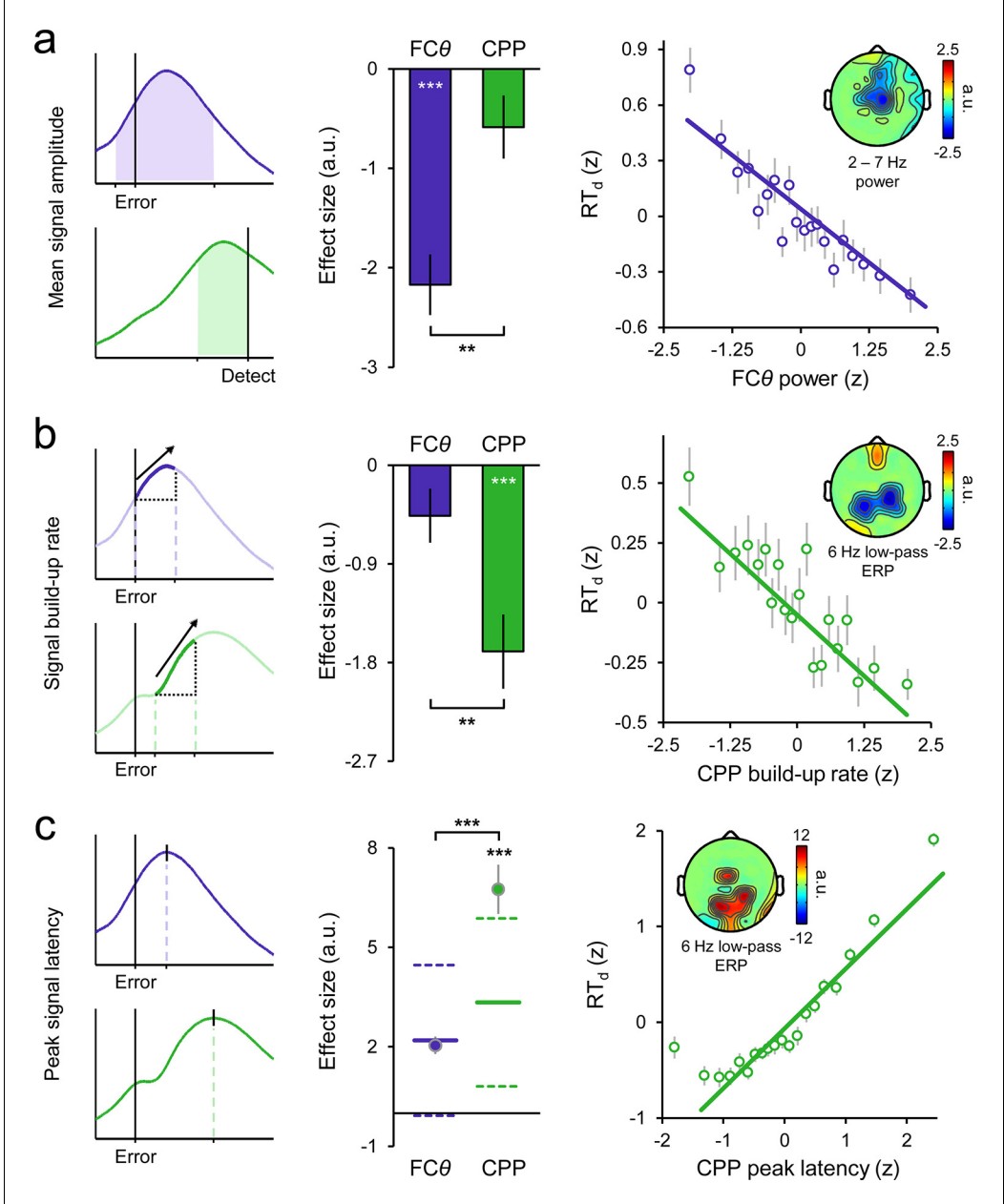

**Figure 4.** Variance in the timing of error detection reports is explained by distinct single-trial metrics of FC$\theta$ and CPP morphology. (a) Schematic depicting the single-trial measurement windows for FC$\theta$ power (purple) and CPP amplitude (green; *left*). Bar graphs (*middle*) show estimated effect sizes from within-subjects robust multiple regressions of RT$_d$ on both amplitude metrics (error bars = s.e.m.), and the associated topography (*inset right*) indicates the scalp distribution of the theta power effect. Scatterplot (*right*) illustrates the linear relationship between FC$\theta$ power and RT$_d$; points and error bars are mean ± s.e.m. of data that were z-scored within subjects, pooled across subjects and grouped into 20 five-percentile bins. (b) Measurement approach, regression-estimated effect sizes, topographic distribution and scatterplot representing the single-trial relationships between RT$_d$ and signal build-up rates. (c) Similar plots representing the relationships between RT$_d$ and peak signal latencies. Points in the effect size plot (*middle*) indicate observed effect sizes, solid and dashed lines highlight the mean and 95% confidence intervals, respectively, of permuted distributions used for significance testing (see *Materials and methods*). Topographies are thresholded at p < 0.005 (a, b) and p < 0.0001 (c) for effect visualization. The solid lines in all scatterplots are simple linear robust regression fits to the unbinned data. **p < 0.01; ***p < 0.001.

The following figure supplement is available for figure 4:

**Figure supplement 1.** Scatterplots illustrating the linear relationships between RT$_d$ and second-order CPP amplitude (a), FC$\theta$ build-up rate (b) and FC$\theta$ peak latency (c).

the CPP did (compare *Figures 2d* and *3c*). Thus, internal fluctuations in FCθ power that were independent of the first-order evidence accumulation process reflected in the CPP were predictive of subsequent error detection behavior. Second, we tested whether the single-trial relationship between FCθ and $RT_d$ (*Figure 4a*) was formally mediated by a direct effect of FCθ on the rate of second-order evidence accumulation by constructing a three-variable path model with FCθ power as the predictor, $RT_d$ as the outcome and CPP build-up rate as the mediator variable (see *Materials and methods*). FCθ was a reliable predictor of CPP build-up rate in this model ($p = 0.0007$) and the mediation effect was significant ($p = 0.0009$), indicating that the rate of second-order evidence accumulation partially mediated the observed relationship between medial frontal signaling and the speed of error detection (*Figure 5*).

## Drift diffusion modelling of the second-order decision process

Informed by the above observations, we modeled error detection as a one-choice diffusion process (*Ratcliff and Van Dongen, 2011*) by which noisy evidence that an error has been committed is accumulated over time (at mean drift rate *v* with between-trial standard deviation *η*). Error detection is achieved in the model once the evidence tally passes a threshold (*a*) whereas the temporal integration process terminates if this threshold is not reached by a time deadline, thereby resulting in an undetected error (*Figure 6a*; *Materials and methods*). The decision to model second-order performance in isolation was based on the observation that higher-order signals (FCθ) play a prominent role in determining second-order decision-making, thus indicating that the input to the primary decision process is not necessarily the same as the input to the second-order process (see *Discussion*). For simplicity, we here focused on decomposing error detection behavior in isolation. This approach served to further identify our electrophysiological signals with distinct features of post-commitment evidence accumulation (evidential input, temporal integration) without requiring speculative assumptions about the relationship between first- and second-order decision processes.

The model fit the error detection data well, capturing the shapes of the group-level and single-subject $RT_d$ distributions (*Figure 6b*; *Table 1*; *Figure 6—figure supplement 1*) as well as the considerable heterogeneity in individuals' capacities for accurate second-order evaluation (*Figure 6c*).

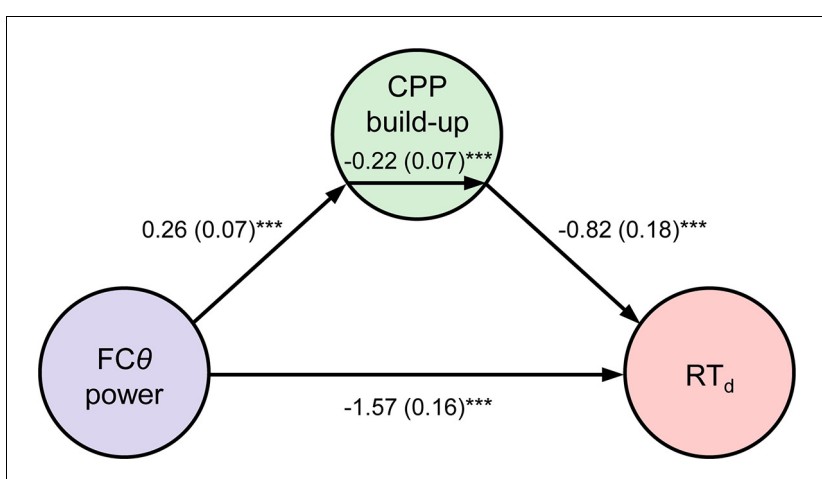

**Figure 5.** Mediation analysis. Path diagram depicts the relationships between nodes in a mediation model that tested whether CPP build-up rate mediated the negative relationship between FCθ power and $RT_d$. Lines are labeled with path coefficients, with s.e.m. shown in parentheses. FCθ power (predictor, *left*) predicted CPP build-up rate (mediator, *middle*), which in turn predicted $RT_d$ (outcome, *right*) controlling for FCθ power. The *upper-middle* coefficient indicates the formal mediation effect. The significant direct path between FCθ power and $RT_d$, calculated controlling for the mediator, indicates partial mediation: CPP build-up rate did not explain all of the shared variance between FCθ and $RT_d$. ***$p < 0.001$, bootstrapped.

The following figure supplement is available for figure 5:

**Figure supplement 1.** Effects of trial binning on CPP build-up rate mediation effect.

**Table 1.** Parameter estimates and goodness-of-fit of error detection diffusion model.

| | a | $t_{nd}$ | v | η | $\chi^2$ |
|---|---|---|---|---|---|
| Mean | 0.21 | 0.20 | 0.45 | 0.45 | 2.46 |
| s.d. | 0.12 | 0.08 | 0.30 | 0.30 | 2.33 |

$\chi^2$ degrees of freedom = 1, critical value = 5.024.

24 of 28 $\chi^2$ values were below the critical value.

We then used the best-fitting model parameters for each subject to generate simulated second-order decision variable trajectories for detected and undetected errors. Despite minimal data-driven constraint on the simulation process (see *Materials and methods*), the temporal evolution of the simulated time-series closely traced that of the second-order CPP (*Figure 6d*). By allowing a proportion of trials to assume negative drift rate, the model additionally provides a parsimonious account of the apparent downward trajectory of the CPP in the averaged waveforms after undetected errors.

The fitted model parameters were also correlated with key FC$\theta$ and CPP signal characteristics across subjects. We employed the per-subject 'drift ratio' ($v/\eta$) as a model-based estimate of each individual's error evidence strength (*Ratcliff and Van Dongen, 2011*; *Materials and methods*) and found that this quantity was positively correlated with FC$\theta$ power ($r = 0.51$, $p = 0.007$; *Figure 6e*). A partial correlation analysis verified that this effect was still present ($r = 0.52$, $p = 0.007$) when $\theta$ power over bilateral posterior electrodes (P7/P8) was included as a covariate, indicating that it is not driven by spurious global differences in oscillatory power (*Cohen and Donner, 2013*). Moreover, a multiple regression to quantify the independent contributions of FC$\theta$ and second-order CPP amplitude to variance in drift ratio revealed a significant effect only for the former ($\beta_{FC\theta} = 0.47$, $p = 0.013$; $\beta_{CPP} = 0.21$, $p = 0.24$). Conversely, drift ratio was correlated with the build-up rate ($r = 0.47$, $p = 0.013$; *Figure 6f*) and peak latency ($r = -0.52$, $p = 0.005$; *Figure 6g*) of the second-order CPP, whereas no such relationships were observed for FC$\theta$ build-up rate ($\beta_{FC\theta} = -0.12$, $p = 0.53$; $\beta_{CPP} = 0.46$, $p = 0.017$) or latency ($\beta_{FC\theta} = -0.10$, $p = 0.64$; $\beta_{CPP} = -0.46$, $p = 0.042$). We also note that neither FC$\theta$ power nor the build-up rate nor peak latency of the second-order CPP were correlated across subjects with primary task behavior (withhold accuracy and primary RT on detected error trials; all $p > 0.1$), thus highlighting the particular sensitivity of these metrics to second-order decision-making. Additionally, these electrophysiological measures were not correlated with other parameters of the computational model (all $p > 0.1$).

## Second-order decision signals are independent of error reporting requirements

Several lines of evidence indicate that the close relationship between the second-order CPP and error detection signalling cannot be attributed to motor preparation or execution. First, we have previously shown that the CPP that precedes a first-order perceptual decision is fully dissociable from motor preparation signals and is observed even when no overt decision-reporting action is required (*O'Connell et al., 2012*). Second, we observed no change in signal topography between the pre- and post-choice phases (*Figure 2d*). Third, the temporally extended and variable build-up of the second-order CPP excludes the possibility that our results can be attributed to the presence of overlapping motor execution potentials. Finally, we asked a new cohort of 12 subjects to perform the same Go/No-Go task but gave no instructions to report errors on half of the task blocks. FC$\theta$ and the CPP were clearly observed following these 'no-report' errors and, consistent with the fact that no-report blocks contain a mixture of detected and undetected errors, linear contrasts confirmed that FC$\theta$ and second-order CPP amplitudes were intermediate between their amplitudes following detected and undetected errors in the condition with self-initiated error reporting (FC$\theta$, $t_{11} = 4.2$, $p = 0.002$; CPP, $t_{11} = 6.9$, $p < 1 \times 10^{-4}$; *Figure 7*).

## Discussion

Our electrophysiological and model-based analyses demonstrate that neural evidence accumulation continues after decision commitment to facilitate reflections on choice accuracy. This empirical

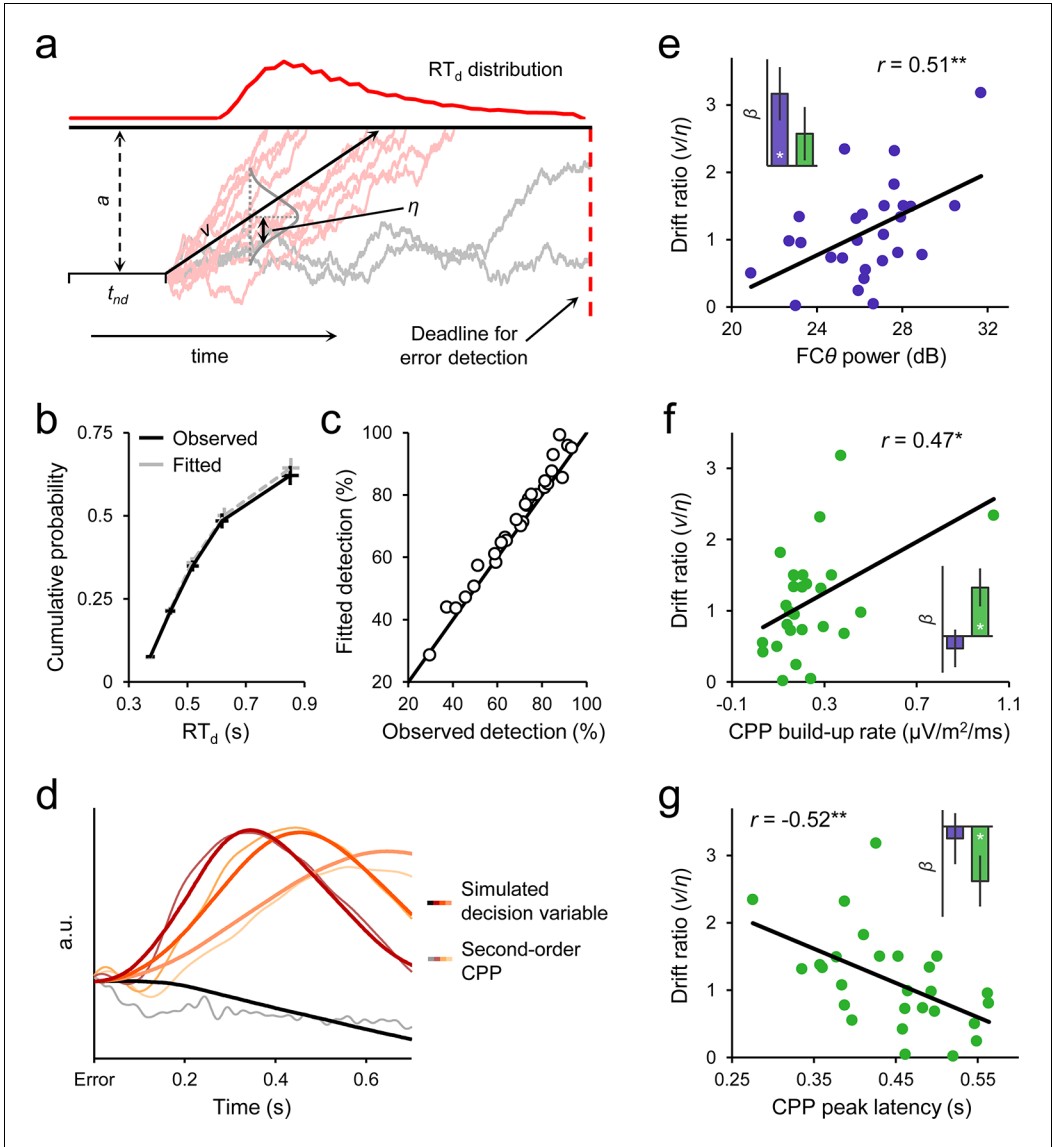

**Figure 6.** Diffusion modelling of error detection behavior. (**a**) Schematic representation of the one-choice drift diffusion model. Noisy error evidence accumulates over time at mean drift rate *v* until a threshold *a* is reached (light red traces) or a deadline on detection expires (gray traces). Drift rate is normally distributed across trials with standard deviation $\eta$, and non-decision-related processing time is captured by $t_{nd}$. Within-trial noise *s* is fixed at 0.1 (not shown). (**b**) Group-level model fit. Points from left to right represent the 0.1, 0.3, 0.5, 0.7 and 0.9 $RT_d$ quantiles, estimated from the data (black) and generated by the model fit (gray). (**c**) Observed versus fitted error detection accuracy. Diagonal line is the identity line. (**d**) Group-average decision variables reflecting the accumulation of error evidence, simulated using the best-fitting model parameters for each subject and overlaid on the grand-average second-order CPP signals. Detected error traces are sorted by $RT_d$ into three equal-sized bins. CPPs are baselined to the 50 ms preceding error. (**e**) Between-subjects relationship between FC$\theta$ power and model-estimated drift ratio ($v/\eta$). Bar graph (*inset*) shows standardized regression coefficients from a multiple regression that included CPP amplitude as an additional predictor. (**f, g**) Scatterplots and bar graphs highlighting between-subjects relationships between drift ratio and signal build-up rates (f) and peak latencies (g). Although one extreme data point is apparent in f, this did not exert disproportionate influence over the fitted regression line (studentized deleted residual < 1) and the correlation remained marginally significant when it was removed (*r* = 0.37, *p* = 0.06). Error bars = s.e.m. *$p$ < 0.05; **$p$ < 0.01.

The following figure supplements are available for figure 6:

**Figure supplement 1.** One-choice diffusion model fits for single subjects.

**Figure supplement 2.** Between-subjects relationship between FC$\theta$ power and model-estimated drift ratio ($v/\eta$) when a trial-averaged baseline was applied to FC$\theta$.

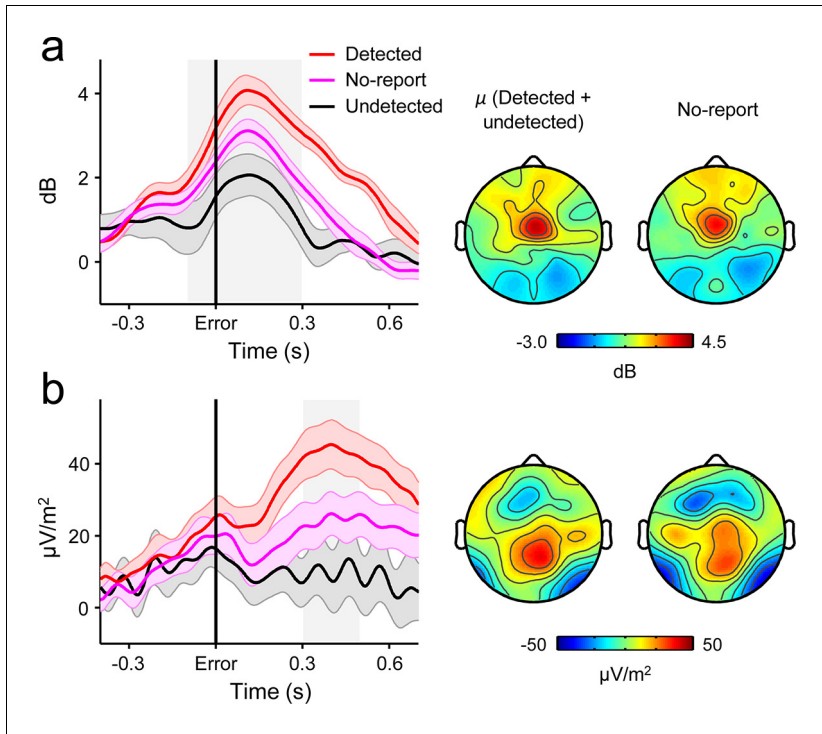

**Figure 7.** Second-order decision signals persist in the absence of error reporting demands. (**a**) Time-courses and topographic distributions of FC$\theta$ power from a new cohort that performed half of task blocks without any explicit instruction to self-monitor performance ('no-report' blocks). Left topography is the average topographic distribution of $\theta$-power after pooling all detected and undetected error trials from blocks with self-initiated error detection reporting; right topography is $\theta$ distribution averaged across all errors in no-report blocks. (**b**) Time-courses and topographies of the second-order CPP component from the same cohort. Conventions are the same as in (**a**). Shaded gray areas in *left* show latencies of associated scalp topographies. Shaded error bars = s.e.m.

observation has important implications for the ongoing debate on the mechanistic basis of metacognition.

A prominent hypothesis is that confidence judgments are based on internal information available at the time of choice commitment (*Heath, 1984*; *Kiani et al., 2014*; *Kiani and Shadlen, 2009*; *Link, 2003*), and models that incorporate this assumption provide a good account of behavior when choice and confidence are interrogated simultaneously. But the manner in which metacognitive evaluations are probed is likely to have a profound impact on how they are constructed in the brain. Whether participating in a laboratory experiment or everyday activity, human decision-makers are most often required to express or act upon their metacognitive evaluations some time after an initial choice. Our results reveal that if the decision-maker is allowed to report delayed, self-initiated judgments of their own performance, these judgments will incorporate new internal information available after the point of commitment, even in the absence of continued sensory input. This process of post-decision evidence accumulation has already been invoked by several computational models to successfully account for delayed confidence judgments and changes-of-mind (*Moran et al., 2015*; *Pleskac and Busemeyer, 2010*; *Resulaj et al., 2009*; *Yu et al., 2015*), while a parallel literature on error detection has long debated the mechanistic basis of post-commitment information processing (*Rabbitt and Vyas, 1981*; *Rabbitt, 1966*; *Yeung et al., 2004*; *Yeung and Summerfield, 2012*). Our findings, however, represent the first definitive neurophysiological demonstration of post-decision evidence accumulation. In so doing, we show that the critical neural dynamics that give rise to both first- and second-order decisions are captured by a single brain signal, opening up new avenues for basic and clinical investigations.

Comparison of pre- and post-decisional CPP dynamics highlighted an important qualitative distinction between the first- and second-order decision processes invoked by our paradigm. The first-

order task required subjects to discriminate between the two possible stimulus categories (i.e. Go versus No-Go), a process that can be understood in terms of competitive accumulation of evidence in favor of each choice alternative (*Gold and Shadlen, 2007*; *Gomez et al., 2007*). We have previously shown that the CPP builds in response to evidence favoring each of the choice alternatives in such contexts (*Kelly and O'Connell, 2013*; *O'Connell et al., 2012*) and accordingly, we observed here that the same signal exhibited a gradual build-up prior to both Go and No-go choices. The second-order task, by contrast, was more comparable to simple signal detection with detected errors translating to hits and undetected errors translating to misses. Evidence accumulation mechanisms have also been invoked to account for signal detection decisions (e.g. *Deco et al., 2007*; *Donner et al., 2009*) but in this case the neural decision variable seems to increase towards a single detection boundary and misses occur when this boundary is not reached (*Carnevale et al., 2012*; *Deco et al., 2007*). Again, previous studies have shown that the CPP in such contexts exhibits a clear ramp-up to threshold for hit decisions that is diminished or absent on miss trials (*O'Connell et al., 2012*), and presently this effect was recapitulated for error detection decisions. Collectively, these findings indicate that human decision-makers are capable of quickly engaging multiple decision processes in succession that, although reliant on the same neural mechanism of evidence integration, are constructed in qualitatively distinct ways.

Our results also offer important new insights into the nature of the input to the metacognitive decision process. Several features of our results mark pMFC activation, as indexed by the power of FC$\theta$ oscillations, as a likely candidate for furnishing a source of this input. First, peri-response FC$\theta$ power was strongly predictive of subsequent error detection. Second, FC$\theta$ power predicted error detection speed from a particularly early latency relative to the CPP. Third, this predictive capacity was formally mediated by the effect of FC$\theta$ on the rate of second-order evidence accumulation, as indexed by CPP build-up rate. Fourth, FC$\theta$ correlated across-subjects with a model-based estimate of the strength of the evidence that fed into the error detection process. This specification of an important role for pMFC signaling in explicit error detection suggests that metacognitive decisions are not solely based on the accumulation of sensory evidence but are also influenced by internally generated higher-order signals.

While these observations accord with theoretical proposals that the pMFC provides a critical input to the error detection process (*Botvinick et al., 2001*; *Yeung et al., 2004*), they are not decisive about the specific nature of this input. One possibility is that pMFC furnishes a distinct source of abstracted 'error evidence' that directly informs second-order judgments. Extensive data suggest that $\theta$-band activity in pMFC signals the degree to which competing action plans are simultaneously activated, commonly known as 'conflict' (*Cavanagh and Frank, 2014*; *Cavanagh et al., 2011*; *Cohen, 2014*; *2013*). Computational models have shown that a post-commitment conflict signal alone would provide a reliable basis for error detection because its magnitude scales with error likelihood and can reliably classify trial-to-trial performance accuracy (*Yeung et al., 2004*; see also *Charles et al., 2014*). Such a causal role for pMFC-encoded conflict in second-order decision-making would also provide a mechanistic explanation for the recent finding that manipulating effector-specific premotor activity both before and immediately after perceptual choice has clear effects on subsequent confidence reports (*Fleming et al., 2015*). Alternatively, second-order decisions on our task may receive their sole or primary evidence from visual short-term memory (*Smith and Ratcliff, 2009*) and pMFC may serve to modulate this process by strategically tuning processing in a global network of task-relevant regions when conflict is detected (*Cavanagh and Frank, 2014*; *Danielmeier et al., 2011*; *Dehaene et al., 1998*; *Shenhav et al., 2013*). Consistent with such a non-specific influence on the post-decisional process, we found that the mediation of the FC$\theta$/RT$_d$ relationship by CPP build-up rate was partial, indicating that pMFC may have also influenced the speed of error detection independently of its effect on the rate of evidence accumulation – perhaps via effects on other parameters of the decision process like the response threshold (*Cavanagh et al., 2011*) or motor execution time. The extent to which pMFC signaling directly modulates the second-order evidence accumulation process can be established in future work by examining the impact of pMFC perturbation on CPP dynamics (*Hayward et al., 2004*; *Reinhart and Woodman, 2014*; *Sela et al., 2012*).

In contrast to other sequential sampling accounts of confidence judgments and changes-of-mind which assume that first-order decisions and metacognitive judgments are both based on the same evidence source (*Moran et al., 2015*; *Pleskac and Busemeyer, 2010*; *Resulaj et al., 2009*;

*Yu et al., 2015*), our simple diffusion model of error detection behaviour is agnostic to the nature of the evidence that drives the second-order decision process. The model thus accommodates suggestions that pMFC, or indeed other neural signals, provide modulatory inputs or additional sources of evidence that are multiplexed in a compound decision variable that determines second-order judgments (*Ullsperger et al., 2014*; *2010*). Of course, a corresponding limitation of our modelling framework is that it does not specify the precise nature of the relationship between the first- and second-order decision processes, and thereby fails to provide explicit accounts of some aspects of our observed data. For example, our selective modelling of error detection behaviour does not provide an explanation for the similarity of the post-response CPP signals on undetected errors and correct Go trials. This similarity could perhaps be due to a common paucity of error evidence on both trial-types. On the other hand, our analysis did reveal a post-response difference in frontal ERP morphology between these trial-types (*Figure 3—figure supplement 1*) which suggests that they are at least partially dissociable in terms of neural dynamics. Such unresolved questions emphasize that a central goal of future research must be to build a unified model of decision-making that not only accounts for the often complex relationships between first- and second-order behavior (*Moran et al., 2015*; *Pleskac and Busemeyer, 2010*), but is also constrained by neurophysiological characterizations of the post-decision accumulation process. Moreover, a remaining challenge will be to devise innovative ways to manipulate the evidence that feeds into the second-order process in an effort to test such a model and further corroborate our identification of the CPP with post-decisional accumulation.

A substantial literature has already investigated error-related electrophysiological signals in human subjects and highlighted in particular a post-response centro-parietal ERP, labeled the Error Positivity (Pe), that reliably discriminates between detected and undetected errors (*Murphy et al., 2012*; *Nieuwenhuis et al., 2001*; *O'Connell et al., 2007*; *Overbeek et al., 2005*; *Ridderinkhof et al., 2009*; *Wessel et al., 2011*). However, a consensus regarding the precise functional significance of this signal has never been achieved. Although a series of recent studies have demonstrated that Pe amplitude correlates with confidence in perceptual decisions (*Boldt and Yeung, 2015*) and the criterion that subjects impose on error detection reports (*Steinhauser and Yeung, 2010*) and have thereby associated this component with the general quality of the metacognitive decision process [see also *Steinhauser and Yeung (2012)*], these studies could not identify the Pe with a specific neural mechanism. The present study, by contrast, demonstrates that the post-decisional CPP encodes a second-order decision variable that bears precisely the same dynamical properties as have been reported for first-order decision signals in single-unit and population-level neurophysiology, including an RT-predictive build-up rate and a boundary-crossing relationship to response execution – properties that have never been previously reported for the Pe. Our use of a self-initiated error detection report was a critical design feature in this regard because it facilitated the interrogation of signal dynamics leading up to the moment of error detection; this has not been possible in previous studies of the Pe, which enforced either delayed metacognitive reporting or none at all. Where we have previously demonstrated that the pre-decision build-up of the CPP encompasses activity that has traditionally been associated with the classic P300 or 'P3b' (*Kelly and O'Connell, 2013*; *O'Connell et al., 2012*; *Twomey et al., 2015*), an important implication of the present findings is that the post-decision build-up of the CPP corresponds to the activity commonly attributed to the Pe. Thus, our data suggest that the P3b and Pe reflect distinct stages of the same neurophysiological process and point to a unifying mechanistic framework for understanding both signals.

In conclusion, we have reported the first definitive neurophysiological demonstration that evidence accumulation continues after the point of decision commitment and predicts the timing and accuracy of subsequent error detection. This finding furnishes critical neurophysiological support for theoretical accounts of metacognitive decision-making that have relied on the concept of post-decisional accumulation. Moreover, we have shown that this process is informed by a higher-order neural signal generated in medial frontal cortex, which suggests that metacognitive judgments are not solely based on the feedforward accumulation of sensory evidence but also on representations of conflict or error likelihood. Collectively, these results shed significant new light on the generative mechanisms of metacognition and furnish new evidence that error detection, confidence judgments and their neural substrates can be understood in terms of the same mechanistically principled framework of evidence accumulation.

# Materials and methods

## Subjects

All subjects were right-handed, had normal or corrected-to-normal vision, no history of psychiatric illness or head injury, reported no color-blindness, and refrained from ingesting caffeine on the day of testing. They provided written informed consent, and all procedures were approved by the Trinity College Dublin ethics committee and conducted in accordance with the Declaration of Helsinki. Subjects received a gratuity of €20 for their participation.

## Task procedures

Testing was performed in a dark, sound-attenuated room. Stimuli were presented using the 'Presentation' software suite (NeuroBehavioural Systems, San Francisco, CA) and subjects responded with the thumb of their right hand using a Microsoft 'Sidewinder' controller. During task performance, subjects used a table-mounted head-rest which fixed their distance from the display monitor (51-cm CRT operating at 85 Hz) at 80 cm for the entire task. They were instructed to maintain gaze at a centrally-presented white fixation cross on a gray background. Color/word stimuli appeared 0.25° above fixation.

## Task version 1: self-initiated error detection reporting

The first version of the task was administered to thirty-two individuals, four of whom were excluded from all analyses: one due to technical issues with the EEG recording, two with excessively poor task accuracy (<30% withheld No-Go trials), and a further subject with no observable CPP component. Thus, we analyzed a final sample of 28 subjects (13 male) for the primary study, with a mean age of 23.5 years (s.d. = 5.8). This pre-planned sample size is consistent with other electrophysiological studies of decision-making from our lab that interrogated similar neural signals and invoked similar analytical methods (*Kelly and O'Connell, 2013*; *O'Connell et al., 2012*; *Twomey et al., 2015*).

Subjects were required to respond with a single 'A' button press on all Go trials, and to withhold this response on both 'repeat' and 'color' No-Go trials (*Figure 1*). They were instructed to give equal emphasis to the speed of their Go responses and the accuracy of their No-Go withholding. In the event of any failure to withhold the pre-potent response to either type of No-Go stimulus, subjects were required to signal detection of this error as quickly as possible by pressing a second 'B' button. They were instructed to execute this error detection report even if they became aware of an error after the onset of the following stimulus. Although such late error detections were rare (mean = 2.5 trials, s.d. = 2.4) and excluded from analysis, this instruction mitigated against the adoption of a time-dependent decision criterion that may have obscured any threshold-like relationship between second-order decision signals and the latency of error detection.

Each subject first completed a brief automated training protocol (*Murphy et al., 2012*), and was then administered at least 8 blocks of the task. Where time constraints allowed, we administered more blocks to increase the number of error trials available for analysis. On average, subjects completed 9.5 ± 0.8 blocks (range 8–10). Each block consisted of 224 word presentations, 200 of which were Go stimuli and 24 of which were No-Go stimuli (12 repeat, 12 color). All stimuli were presented for 0.4 s, followed by an inter-stimulus interval of 1.6 s. The duration of each block was therefore approximately 7.5 minutes. Stimuli were presented in a pseudo-random order with a minimum of three Go trials between any two No-Go trials.

## Task version 2: half task blocks without error detection reporting

To establish the generality of the post-decisional signals observed under the previous task version to situations in which decision-makers are not explicitly instructed to monitor their own performance, we tested an independent cohort of sixteen individuals on a version of the Go/No-go task in which error reporting instructions were manipulated. Four of these subjects were excluded from analysis due to insufficient numbers of undetected errors (<6) for reliable EEG analysis, leaving a final sample of twelve (5 male) with a mean age of 23.7 years (s.d. = 6.7). Although this sample size is lower than that employed in the first study, we here limited our analyses to dependent measures with inherently high signal-to-noise (trial-averaged waveforms) and this sample size is consistent with a previous study of error detection using the same task employed presently (*Hester et al., 2005*). Task design

and procedures were identical to those used for the previous version of the task, with the exception that every subject was administered five task blocks with regular error detection reporting and five 'no-report' blocks in which subjects were not instructed to monitor for errors. The five blocks in each condition were administered contiguously, and the order in which each condition was presented was counter-balanced across subjects. Thus, half of the full sample performed the no-report blocks without knowing that they would later be required to signal their errors. The presence of both FC$\theta$ and CPP signals in the no-report condition (*Figure 7*) did not depend on condition order.

## EEG acquisition and preprocessing

Continuous EEG was acquired using an ActiveTwo system (BioSemi, The Netherlands) from 64 scalp electrodes, configured to the standard 10/20 setup and digitized at 512 Hz. Eye movements were recorded using two vertical electro-oculogram (EOG) electrodes positioned above and below the left eye and two horizontal EOG electrodes positioned at the outer canthus of each eye. EEG data were processed in Matlab (Mathworks) via custom scripting and subroutines from the EEGLAB toolbox (*Delorme and Makeig, 2004*).

Eye-blinks and other noise transients were isolated and removed from the EEG data via independent component analysis (ICA). Specifically, continuous data from each block were re-referenced to channel Fz; data were high-pass filtered to 1 Hz, low-pass filtered up to 95 Hz and notch filtered to remove 50 Hz line noise using a two-way least-squares FIR filter; noisy channels were identified by visual inspection of signal variance and removed; data were segmented into temporally contiguous epochs of 1 s duration; epochs containing values that violated amplitude (± 250 μV) and joint probability (± 4.5 s.d.; *Delorme and Makeig, 2004*) criteria were rejected; and, the remaining data were subjected to temporal ICA using the infomax algorithm. The ICA weights yielded by this procedure were then back-projected to the original continuous, unfiltered EEG data for the associated block. Next, independent components representing stereotyped artifactual activity such as eye blinks, saccades and individual electrode artifacts were identified by visual inspection and discarded, and the ICA-pruned data were low-pass filtered to 35 Hz. No high-pass filter was applied. Previously-identified noisy channels were then interpolated via spherical spline interpolation, and the data were re-referenced to the common average. Data epochs were extracted from 2.5 s before to 3.5 s after stimulus onset on each trial (thus minimizing edge artifacts during spectral analysis) and baseline-corrected to the 0.3 s interval preceding stimulus onset. Subsequent epoch rejection employed a dynamic window with a fixed start time of −0.3 s relative to stimulus onset and an end time that depended on the primary RT of each trial: the window ended at RT + 1.2 s for go trials and undetected errors, and RT + 0.2 s + the slowest RT$_d$ of the current participant for detected errors; end time was truncated to +2 s relative to stimulus onset if the window encroached upon onset of the subsequent stimulus on any trial. Epochs were rejected from all further analysis if any scalp channel exceeded ±100 μV at any point within this trial-specific window. Detected error trials on which RT$_d$ followed next-trial onset were also excluded. Lastly, all EEG data were converted to current source density (*Kayser and Tenke, 2006*) to increase spatial selectivity and minimize volume conduction (*Kelly and O'Connell, 2013*; *Twomey et al., 2015*).

A previous paper by our group investigated the relationship between the Pe component (referred to here as the second-order CPP) and error detection using the same data and reported a strong correlation between Pe peak latency and RT$_d$ (*Murphy et al., 2012*). However, we did not consider the influence of Pe build-up rate on performance, we did not establish functional equivalence between the Pe and CPP and we did not consider the contribution of FC$\theta$ to error detection (see below). In fact, in that paper we reported a reliable negative association between RT$_d$ and Pe amplitude immediately prior to error detection that is inconsistent with the presently reported results and difficult to reconcile with the proposal that the Pe reflects the accumulation of evidence toward a fixed decision bound. Two critical methodological distinctions between Murphy et al. and the present study account for this discrepancy. First, whereas Murphy et al. implemented a high-pass temporal filter, the present study did not. High-pass filtering is problematic in the current context because it is likely to attenuate CPP amplitude to a greater degree on trials with long RTs since decision-related neural activity is drawn out over a longer time frame on such trials. Second, the present study used a spatial filter to reduce the overlap with other temporally coincident signals because in recent work we demonstrated that boundary-crossing effects at response can be obscured by spatial overlap of the CPP with anticipatory signals emanating from frontal sites (*Kelly and O'Connell, 2013*;

*Twomey et al., 2015*). Addressing these issues enabled us to make a variety of important new observations that extend our previous results in several significant ways: we provide the first demonstration of the critical build-to-threshold relationship between the second-order CPP and $RT_d$ that is characteristic of an evolving decision variable; we show that this component interacts with a frontal conflict signal to determine the probability and timing of error detection; and, we leverage computational modelling to further identify this component with the second-order evidence accumulation process.

Fronto-central theta (FC$\theta$; 2–7 Hz) power was measured in two ways. First, EEG data were decomposed into their time-frequency representation via complex Morlet wavelet convolution (between 2 and 12 cycles per wavelet, linearly increasing across 90 linear-spaced frequencies from 1 to 30 Hz) and the resulting power estimates were normalized by the decibel (dB) transform (dB = $10*\log_{10}$[power/baseline]). The baseline consisted of across-trial average power during the 0.3 s preceding stimulus onset, calculated and applied separately within each trial-type (Go, undetected error, detected error). This approach yielded a condition-averaged time-frequency plot that allowed us to select channels and frequency boundaries for FC$\theta$ analysis in a manner that was orthogonal to any potential trial-type effects. Black lines in this plot (*Figure 3a*) enclose regions in which contiguous time-frequency pixels were significantly different from the pre-stimulus baseline at $p < 0.01$, for at least 400 ms and at least 5 consecutive frequency bins. Second, power estimates for all subsequent FC$\theta$ analyses were derived by band-pass filtering the EEG data from 2 to 7 Hz (using the *fir1* Matlab function to construct a narrow two-way least squares FIR filter kernel), Hilbert-transforming the filtered data to derive the analytic signal, and then converting to power. Compared to wavelet convolution this filter-Hilbert method affords greater control over the frequency characteristics of the filter, though in practice the results from both methods were qualitatively very similar. As for the previous wavelet-based approach, analyses were conducted on power estimates that were dB-normalized using a condition-specific trial-averaged pre-stimulus baseline, thus leaving within-condition trial-by-trial fluctuations in baseline power intact. Complementarily, the reported between-subjects FC$\theta$ correlation (*Figure 6e*) was conducted on unbaselined power estimates, thus leaving between-*individual* differences in baseline FC$\theta$ power intact. Both within- and between-subjects variation in baseline power emerged to be sources of variance contributing to the respective effects (*Figure 3—figure supplement 3c-e*; *Figure 6—figure supplement 2*). We also verified that the main effect of error detection on FC$\theta$ remained unchanged when no baseline was applied (*Figure 3—figure supplement 3b*).

## Analysis of electrophysiological decision signals

First- and second-order trial-averaged CPP signals were measured as the average voltage per m$^2$ from three centro-parietal electrodes centered on the region of maximum component amplitude in the grand-average response-locked topography (Pz, P1, P2), and were low-pass filtered up to 10 Hz for analysis and display. FC$\theta$ was measured as the average power across six fronto-central electrodes, also centered on the topographic maximum (FCz, FC1, FC2, Cz, C1, C2). The relationships between RT and signal build-up rates and amplitudes were examined for the CPP on go trials (*Figure 2a*), the second-order continuation of this signal on detected errors (*Figure 2d*), and FC$\theta$ on both trial-types (*Figure 3d,e*). Analysis of Go-trial dynamics was restricted to trials with primary RTs > 350 ms (leading to the exclusion of 17.4 ± 12.1% of trials per subject) because the amplitude of the CPP signal was affected by visual-evoked potentials that coincided with the evolution of the decision-related activity on quicker trials (*Figure 2—figure supplement 2*). For each participant, single-trial waveforms were sorted into 3 equal-sized bins according to RT (primary RT for Go trials, $RT_d$ for detected errors) and averaged. To establish the timing of the relationship between signal build-up rate and RT, we measured the temporal slope of each signal in each subject's bin-averaged waveforms using a sliding window of 150 ms, covering the entirety of both response- and detection-aligned waveforms. Build-up rate was computed as the slope of a straight line fitted to the signal within each temporal window, and a linear contrast was applied to this metric across RT bins. The centers of windows that were characterized by a significant group-level contrast in the expected direction (linear decrease in build-up rate with increasing RT; $p < 0.05$, one-tailed) are marked by a black running line in the associated plots. To establish the timing of the relationship between signal amplitude and RT, we conducted a linear contrast of amplitude as a function of RT bin for each temporal sample. Samples characterized by contrasts that deviated from zero at the group level ($p <$

0.05, two-tailed) are marked by a gray line in the associated plots. This sample-wise approach was also employed to characterize the effect of error detection on the amplitude of both centro-parietal (*Figure 2b*) and FC$\theta$ (*Figure 3b*) signals.

For FC$\theta$ and CPP ROC curve analyses (*Figure 2c*; *Figure 3c*), single-trial waveforms for detected and undetected errors were pooled across all subjects and the area under the ROC curve was calculated for the average of each signal within discrete peri-error time windows (window width of 20 ms, moving in 20 ms increments, from −0.4 to +0.6 s relative to error). Significant deviations in classification accuracy from chance levels were determined via permutation testing (1000 iterations with random trial reassignment conserving individual detected versus undetected error proportions).

All further single-trial analyses of the second-order CPP leveraged waveforms that were low-pass filtered to 6 Hz to increase signal-to-noise. Single-trial amplitude was defined as the mean power from –0.1 to +0.4 s relative to error commission for FC$\theta$ and the mean signal in the 0.2 s preceding error detection for the CPP (*Figure 4a*). Single-trial build-up rate was measured as the slope of a straight line fitted to each waveform using the interval 0 to +0.2 s for FC$\theta$ and +0.1 to +0.3 s for the CPP, both relative to error commission (*Figure 4b*). Single-trial peak latency was measured as the time of maximum signal amplitude relative to error commission within a dynamic measurement window with a start time of −0.1 s for FC$\theta$ and +0.1 s for the CPP, and an end time of the $RT_d$ for that trial +0.15 s (*Figure 4c*). In cases where trials were initially assigned the minimum possible latency given the above constraints, the window start time was adjusted to be the earliest latency at which the waveform next became positive-going.

The independent contributions of FC$\theta$ and the CPP toward $RT_d$ were examined via single-trial within-subjects robust regression (*O'Leary, 1990*). For signal amplitude, the equation $RT_d = \beta_0 + \beta_1 * FC\theta_{Amp} + \beta_2 * CPP_{Amp}$ yielded fitted regression coefficients representing the linear relationships between $RT_d$ and both single-trial theta power ($\beta_1$) and CPP amplitude ($\beta_2$). For build-up rate and peak latency, two further models were constructed by replacing the predictor variables where appropriate. All coefficients in a given model were estimated simultaneously via type III sum of squares. $RT_d$ was log-transformed and peak signal latency was square root-transformed to normalize their respective distributions before coefficient estimation. Variance inflation factors for all predictors across all models were <1.8, indicating weak multi-collinearity. For the amplitude and build-up rate metrics, effect sizes of the fitted $\beta$ coefficients (effect size $t = \beta/s.e.m.$) were tested for group-level statistical significance via one-sample $t$-test ($H_0$: effect size = 0) and contrasted via paired $t$-test ($H_0$: effect size$_1$ − effect size$_2$ = 0). For the latency metric, our use of a dynamic measurement window that was determined by single-trial $RT_d$ ensured an arbitrary positive correlation between single-trial signal latency and RT. To test for effect significance, we thus compared the observed effect sizes derived from the above regression model to the expected values of the same effect size metrics computed in the case in which signal latencies were randomly chosen from anywhere within each trial's measurement window. This process was repeated 1,000 times per subject to derive subject-specific permuted distributions of the FC$\theta$ and CPP latency effects, against which we tested for statistical significance. The above procedures for amplitude, build-up rate and latency effects were also repeated on an electrode-by-electrode basis to construct topographic representations of where these effects were strongest on the scalp (*Figure 4*). For the latency analysis, the trial-by-trial timing of the peak positivity or negativity was extracted for each electrode, depending on whether the trial-averaged amplitude at that electrode in the 0.3 s preceding the error detection report was above or below zero, respectively.

We employed mediation analysis (M3 toolbox for Matlab; http://wagerlab.colorado.edu/tools) to establish whether second-order CPP build-up rate mediated the relationship between FC$\theta$ power and $RT_d$ (*Figure 5*). For this analysis, we measured CPP build-up rate from −0.3 to −0.1 s relative to error detection report in order to minimize the temporal overlap between the FCθ and CPP measures, though the results were very similar when the original error-aligned measurement window for build-up rate was employed. Single-trial values for each measure were z-scored within-subjects and pooled across-subjects. To mitigate the low signal-to-noise ratio inherent in correlating two noisy single-trial electrophysiological metrics, average values for each of FC$\theta$, CPP build-up and $RT_d$ were computed in bins of 12 trials that were grouped after sorting trials in order of increasing $RT_d$. Mediation effects at larger bin sizes were of comparable magnitude (*Figure 5—figure supplement 1*). For CPP build-up rate to be considered a significant mediator, it was required to reach statistical significance in three tests: it must be related to the predictor (FC$\theta$), it must be related to the outcome

(RT$_d$) while controlling for the predictor, and the mediation effect (evaluating whether some covariance between predictor and outcome can be explained by the mediator) must be significant. Significance of the associated path coefficients was assessed via bias-corrected bootstrap tests with 10,000 samples (*Wager et al., 2008*).

## Diffusion modelling

Second-order behavioral data (error detection accuracy and RT$_d$) were decomposed into latent decision-making parameters via a novel application of the one-choice drift diffusion model (*Ratcliff and Van Dongen, 2011*; *Figure 6a*). Noisy evidence for an error was assumed to accumulate over time at drift rate $v$ until a decision bound $a$ was met, at which point error detection was achieved. The moment-to-moment noise in the evidence is determined by the $s$ parameter, which refers to the standard deviation of a zero-mean Gaussian distribution from which random increments to the deterministic component of the accumulation process (represented by $v$) are drawn. As is common in fits of the drift diffusion model to data, $s$ was fixed at 0.1 in order to scale all other parameters in the model across individuals. Drift rate was assumed to be normally distributed across trials with a standard deviation $\eta$, and all non-decision-related processing was assigned to a non-decision time parameter $t_{nd}$. The inclusion of the $\eta$ parameter accorded with previous one-choice model fits to first-order behavior which suggested that variability in drift rate is necessary to account for the various observable shapes of hazard function in one-choice data (*Ratcliff and Van Dongen, 2011*). We made the additional assumption that the temporal integration process terminated if $a$ was not reached by a time deadline, thereby resulting in an undetected error. A deadline on post-decision accumulation also features in a prominent model of fast changes of mind in decision-making (*Resulaj et al., 2009*) and was a free parameter in that study. Here, we estimated the subject-specific detection deadline empirically in order to retain a degree of freedom when assessing model fit: any extreme outliers (> mean + 3.5 s.d.) were trimmed from each subject's RT$_d$ distribution and the deadline was defined as the slowest remaining RT per subject. This procedure yielded an average deadline of 1.17 s relative to initial error commission ($\pm$0.19; range 0.81 to 1.47 s).

Two additional assumptions were made when plotting simulated decision variables derived from the best-fitting model parameters (*Figure 6d*), both of which were informed by characteristics of the second-order CPP. First, 90 ms of each subject's fitted $t_{nd}$ parameter was allotted to post-threshold response preparation, and any residual $t_{nd}$ determined the length of the delay between error commission and the start of evidence accumulation. Second, the decision variable was subject to a linear decay to baseline over the 300 ms after the decision bound was reached.

There is no analytical solution for RT distributions with negative drift rate, which can result from including $\eta$ in the one-choice model (*Ratcliff and Van Dongen, 2011*). The model was therefore implemented as a simulation using a random walk approximation to the diffusion process with 15,000 iterations per distribution at 1 ms step size. In order to fit the model to the observed data for each subject, five RT quantiles (0.1, 0.3, 0.5, 0.7, 0.9) were computed from that subject's RT$_d$ distribution and the proportions of all error trials (detected + undetected) lying between those quantiles were multiplied by the total number of error trials to yield *observed values (O)*. Thus, the *defective* cumulative probability distribution of error detection reports was used to derive per-quantile trial frequencies, which allowed the model to simultaneously fit both RT$_d$ and error detection accuracy. We then calculated the model-estimated proportions of trials that lay between these RT quantiles, and these were multiplied by the number of actual observations to yield the model-derived *expected values (E)*. A $\chi^2$ statistic $\Sigma(O - E)^2/E$ was computed and the parameters of the model were adjusted by a particle swarm optimization routine (*Birge, 2003*) to minimize this value iteratively (30 particles, set at pseudorandom starting points in parameter space). Initial attempts at parameter estimation indicated that the particle swarm approach tended to be more robust to local minima than the commonly-used Simplex minimization routine.

As described elsewhere (*Ratcliff and Van Dongen, 2011*), the one-choice diffusion model suffers from a parameter identifiability problem whereby different estimates of the $v$, $\eta$ and $a$ parameters can produce similar goodness of fit but vary in magnitude by as much as 2:1. However, the ratio of $v/\eta$ remains invariant across different model fits. We therefore employed this 'drift ratio' quantity, which is analogous to $d'$ in classic signal detection theory (*Ratcliff and Van Dongen, 2011*), as our model-based estimate of the quality of the evidence feeding into each subject's error detection decision process. In the reported between-subjects correlations (*Figure 6e,f,g*), drift ratio was

correlated against summary electrophysiological measures for each subject. Amplitude and peak latency measures were calculated by averaging across single-trial estimates of these metrics on detected error trials, derived via the same measurement windows that were used for the previous within-subjects regression analyses. In an effort to increase the signal-to-noise of the build-up rate metric, per-subject build-up rate for each signal was defined as the slope of a linear fit to the average error-locked waveform on detected error trials, within a window that started at a subject-specific signal onset time obtained by visual inspection and ended at the subject-specific peak signal latency. One outlier data point with an absolute studentized deleted residual value > 4 in all bivariate correlations was excluded from the reported analyses, though all relationships remained at least marginally significant when this subject was included (all $p < 0.06$).

## Acknowledgements

This work was supported by an Irish Research Council (IRC) "Embark Initiative" grant (PRM) and a European Research Council (ERC) Starting Grant (ROC) under the European Union's Horizon 2020 research and innovation programme (grant agreement No 638289). The authors thank Simon Kelly and Sander Nieuwenhuis for helpful discussions. The authors declare no conflicts of interest.

## Additional information

### Funding

| Funder | Grant reference number | Author |
| --- | --- | --- |
| Irish Research Council | Postgraduate Fellowship | Peter R Murphy |
| European Research Council | Starting Grant 63829 | Redmond G O'Connell |

The funders had no role in study design, data collection and interpretation, or the decision to submit the work for publication.

### Author contributions

PRM, Conceived and designed the study, acquired data, analyzed and interpreted data, drafted the manuscript, contributed critical revisions, approved the final version; IHR, Conceived and designed the study, contributed critical revisions, approved the final version; SH, Analyzed and interpreted data, contributed critical revisions, approved the final version; RGO'C, Conceived and designed the study, analyzed and interpreted data, drafted the manuscript, contributed critical revisions, approved the final version

### Ethics

Human subjects: Subjects provided written informed consent, and all procedures were approved by the Trinity College Dublin ethics committee and conducted in accordance with the Declaration of Helsinki.

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
