## [Decision Letter]

Thank you for submitting your work entitled "Neural evidence accumulation persists after choice to inform metacognitive judgments" for consideration by *eLife*. Your article has been reviewed by two peer reviewers, one of whom has agreed to reveal their identity: Jim Cavanagh. The evaluation has been overseen by a Reviewing Editor (Michael Frank) and Timothy Behrens as the Senior Editor.

The reviewers have discussed the reviews with one another and the Reviewing editor has drafted this decision to help you prepare a revised submission.

Summary:

This paper reports a novel investigation of the dynamics of post-decisional integration of evidence in error detection. There has been considerable recent interest in the idea that sensory evidence continues to accumulate after a decision, and might underpin metacognitive phenomena such as confidence judgments and changes of mind. So far this notion has only been tested with behavioral modeling, but this paper reveals a possible biological mechanism. The authors present evidence suggesting that the post-error voltage positivity in the ERP (the Pe) reflects the same sort of evidence accumulation process as the central parietal positivity (CPP). This work thus re-defines well-known neuroelectric events based on more generalized and formalized algorithmic definitions. While the authors have previously described the CPP as a first-order evidence (direct sensory evidence) accumulation process, here they argue that the post-error process described here (Pe) reflects second-order (endogenous evidence) evidence accumulation. The authors bolster this claim by applying a one-choice diffusion model to the RT data, facilitating a direct comparison between the mechanistic implementation of CPP-like algorithms (P3 and Pe) even though they have differing cognitive implications (first vs. second order decisions).

Essential revisions:

The methodological sophistication of this report leads to some compelling strengths yet also a few lingering concerns. Virtues of this approach include intelligent spatial and temporal filtering, appropriate capitalization on previously detailed understanding of the role of non-phase locked frontal theta power in post-error decision making (including a formalized mediation model of theta, CPP and RT change), and successful one-choice diffusion modeling and application of model parameters to assist in the generalization of these findings. The covariation of the CPP with the second-order detection response time is particularly clear, providing strong underpinnings for post-decision evidence accumulation frameworks. The concern that the second-order CPP response is a motor artefact is allayed by observation of a similar potential in a "no response" control experiment. But there are some lingering concerns that both reviewers highlighted, elaborated in detail below, consolidating comments from individual reviewers.

1) A limitation of the modelling framework is the omission of any relationship to objective task requirements. The error detection process is modelled as a noisy one-response accumulator with its own drift rate, but this process does not know anything about the first-order decision. I understand the benefit of keeping things simple here and not making unnecessary assumptions about the link between first and second order accumulation, but I worry that it gives a misleading picture of the underlying second order accumulation process, and what we should expect of the data. Specifically, the model predicts that objective errors that don't reach a response threshold are no different, neurally, from correct "go" trials. Instead it seems more plausible that objective features of error trials drive fluctuations in "error evidence" and that on a subset of trials this evidence reaches criterion for report. It's therefore also surprising that the CPP and FCθ signals for both "Go" and "Undetected errors" (which are matched for motor output) were also near-identical (Figure 2 and Figure 3). Doesn't this finding go against the notion that these signals are accumulating noisy evidence to a bound? Or are these signals instead indexing the criterion for error reporting? Perhaps there are differences between undetected errors and go trials in the data that would shed light on this issue that are not apparent from Figure 2 and Figure 3?

2) More generally, caution is warranted on the claims that error detection reflects evidence accumulation per se after the primary task response has been executed. There is no manipulation of "evidence" strength (i.e. where faster accumulation would proceed for larger evidence), and no speed-accuracy trade-off manipulation to test for some kind of bounded accumulation, either of which would provide stronger support that the error detection process reflects a similar sequential sampling of evidence. Thus the authors should discuss and/or address this limitation.

3) Some analytic choices appear to be unrelated to other similar analyses, raising worrying concerns of tailoring the final report to the best outcomes of multiple choices rather than relying on the output of a single analytic framework. This decreases confidence in the robustness and thus the replicability of these full sets of findings and raises concerns of "p-hacking" based on the best of many seemingly arbitrary analytic choices (which I do not think are arbitrary at all). In particular, in paragraph four of the subsection “EEG acquisition and preprocessing” the authors describe the use of differing baseline procedures for frontal theta, which is potentially problematic. As the authors allude to ("leaving within-condition trial-by-trial fluctuations in baseline power intact"), condition-specific pre-stimulus baselines could have important influence on the appropriate level of trial-to-trial vs trial-specific inference, yet it could also simply add spurious variance into the dB ratio of post-event power. The authors do not distinguish between these possibilities or otherwise adequately justify their approach. Moreover, the between-subjects contrasts were not baseline corrected and the authors describe how both are "important source(s) of variance contributing to the reported effects". The authors need to do a more compelling job of justifying their analytic choices based on a principled, a priori framework, for example a common cross-condition baseline for all contrasts (see Cohen 2014: Analyzing Neural Time Series Data, MIT Press).

4) In the subsection “Signal interactions predict the timing of error detection” and in paragraph four of subsection “Analysis of electrophysiological decision signals”: The authors need to specify their regression models in greater detail. They report that "single-trial regressions revealed that this effect was only marginally significant after accounting for the contribution of CPP build up rate". However, it isn't made explicitly clear if they are included in a sequential or type III sum of squares manner to account for variance in main effects prior to the other main effects.

5) Paragraph one of subsection “Analysis of electrophysiological decision signals”: One-tailed test are rarely appropriate for hypotheses that could possibly go in either direction. Here, it is entirely possible that there would be a linear decline in an EEG signal associated with longer RTs.

6) The inference of a causal link between FCθ and CPP seems on shaky ground. Comparing Figure 2 and Figure 3, FCθ power differentiates between RTd earlier than CPP, but could this be due to the influence of neighbouring timepoints on the wavelet measure? This could be addressed with narrow-band wavelets or a filter-HIlbert approach. In single-trial analyses the FCθ power window (-0.1 to +0.4s) extends beyond the window at which the CPP build-up rate is calculated (+0.1 to +0.3s; Methods, paragraph three of the subsection “Analysis of electrophysiological decision signals”). It's therefore odd to enter FCθ as a cause of CPP in a mediation analysis given that the relevant part of the former potentially takes place after the latter. Putting aside the issue of timing, could the assumed causality be further bolstered by flipping the nodes in the mediation analysis, i.e. could the authors show that FCθ does not mediate the influence of CPP build-up on RTd?

7) The first-order CPP increases for both go and no-go trials, consistent with the notion of a domain-general accumulator. But the second-order CPP increases only on error trials (Figure 2). If the second-order decision is analogous to the first (i.e. a race between neural subpopulations controlling whether or not to press the button based on fluctuating evidence), then presumably there should also be accumulation towards a no-go bound in the second-order, confident-in-being-correct case. Doesn't this mismatch between first-order and second-order CPP features weaken the conclusion that the same DV mechanism underpins first- and second-order decisions? It would be interesting to hear the author's perspective on this. For instance the second-order CPP could reflect evidence accumulation from endogenous activities towards an overt error detection response, but the authors also argue the first-order CPP is also involved in evidence accumulation leading to a non-response (Figure 2—figure supplement 1).

[Editors' note: further revisions were requested prior to acceptance, as described below.]

Thank you for resubmitting your work entitled "Neural evidence accumulation persists after choice to inform metacognitive judgments" for further consideration at *eLife*. Your revised article has been favorably evaluated by Timothy Behrens (Senior editor), Reviewing editor Michael Frank, and two reviewers (Steve Fleming and James Cavanagh). All were largely satisfied with your revision, but one Reviewer notes an additional analysis/discussion on the differences between go trials and undetected errors and a supplemental figure on the EEG time courses that would be helpful. Please address these issues and we will then be able to swiftly come to a final official decision.

Reviewer #1:

In my opinion, the findings presented here have been thoroughly vetted. Highly novel work raises many important questions. The authors have done a very reasonable job of fully addressing the 'solvable' issues raised here while also intelligently responding to larger-scale conceptual questions that will require multiple experiments to fully understand.

Reviewer #2:

The authors have mostly addressed my previous concerns in the first round of reviews. The one part that I don't think has been fully addressed is the question of whether there are actually differences between undetected errors and go trials in the data that would not be captured by the model (in general I'm happy with the response to point 1 that the connection between first- and second-order accumulation can be left for future work, but it seems important to fully display the data features that might inform such work). For instance the ERP analysis in Figure 3—figure supplement 1 only directly contrasts undetected and detected errors, but there appears to be a difference between go and undetected errors later in the time course. Similarly only go-trial build-up rates are documented in Figure 2 and Figure 3. I suggest including the comparison between go and undetected CPP and FCθ timecourses as a supplemental figure to cover this issue.

---

## [Author Response]

*1) A limitation of the modelling framework is the omission of any relationship to objective task requirements. The error detection process is modelled as a noisy one-response accumulator with its own drift rate, but this process does not know anything about the first-order decision. I understand the benefit of keeping things simple here and not making unnecessary assumptions about the link between first and second order accumulation, but I worry that it gives a misleading picture of the underlying second order accumulation process, and what we should expect of the data. Specifically, the model predicts that objective errors that don't reach a response threshold are no different, neurally, from correct "Go" trials. Instead it seems more plausible that objective features of error trials drive fluctuations in "error evidence" and that on a subset of trials this evidence reaches criterion for report. It's therefore also surprising that the CPP and FCθ signals for both "Go" and "Undetected errors" (which are matched for motor output) were also near-identical (Figure 2 and Figure 3). Doesn't this finding go against the notion that these signals are accumulating noisy evidence to a bound? Or are these signals instead indexing the criterion for error reporting? Perhaps there are differences between undetected errors and Go trials in the data that would shed light on this issue that are not apparent from Figure 2 and Figure 3?*

We agree that, without modelling first- and second-order performance within a single unified framework, we are unable to conclusively address certain aspects of the presented data. We also accept that the manuscript should do a better job of communicating what our modelling efforts do and do not tell us about the mechanisms underpinning error detection.

We do maintain that our current modelling efforts support the statement that “features of error trials drive fluctuations in ‘error evidence’ and […] on a subset of trials this evidence reaches criterion for report”. The one-choice diffusion model explicitly incorporates such variability in evidence strength (in the form of the s and η parameters, which specify within- and between-trial variability, respectively), and this variability is the primary determinant of whether or not an error will ultimately be detected. Indeed, a key strength of our modelling approach is to show that even though we make very few assumptions beyond this basic feature of sequential sampling models, our one-choice model is able to reproduce both observed error detection behavior (Figure 6) and the associated neural dynamics (Figure 6) – including, of particular note, the negative-going trial-averaged trajectory of the CPP on undetected errors. Thus, the model provides a very parsimonious account of the behavioral and neural error detection data.

On the other hand, strictly speaking our model makes no predictions about the neural equivalence (or otherwise) of undetected errors and Go trials, since behavior and neural dynamics on Go trials were not modelled. The reviewers suggest that the similarity of the neural signals on these trial-types (Figure 2, Figure 3) is inconsistent with the notion that the post-response CPP signal reflects post-decisional accumulation. However, we note that similar trial-averaged neural dynamics on Go and undetected error trials are not necessarily inconsistent with a post-decision accumulation account. Go trials are also likely to be associated with varying levels of error evidence (e.g. on tasks that evoke response conflict, FCθ power has been shown to vary strongly across correct trials and be predictive of task behavior, even within conflict conditions; Cohen & Cavanagh, 2011). Although in most cases the error evidence is likely to be weak, the presence of a minority of trials with elevated error evidence may be sufficient to yield equivalent trial-averaged signals to those observed on undetected error trials, which represent the subset of error trials on which the internal representation of evidence was weakest. This possibility accommodates our current modelling efforts which, as described above, specify that post-decisional accumulation takes place on both undetected and detected error trials. We do generally agree with the reviewers that our findings are not decisive about the nature of the relationship between the first- and second-order decision processes, and that this remains a key area of future inquiry. We have now attempted to further highlight this point in the Discussion section of our manuscript (paragraph six).

The reviewers also question whether the FCθ and/or CPP signals might be better understood as a reflection of the criterion for error reporting. Several aspects of our data speak against this idea. We observed that FCθ power was negatively associated with the timing of error detection (Figure 3, Figure 4) – opposite to the direction that would be predicted if this signal reflects trial-by-trial variation in the second-order criterion (a higher criterion should equate to slower RTs, and hence the relationship would be positive). Additionally, it is unclear to us why a signal like the CPP, which exhibits a clear RT-predictive build-up rate (Figure 4) and whose evolution finely traces variation in the putative evidence accumulation process (Figure 6), might reflect a static criterion as opposed to this more dynamic aspect of the decision process.

*2) More generally, caution is warranted on the claims that error detection reflects evidence accumulation per se after the primary task response has been executed. There is no manipulation of "evidence" strength (i.e. where faster accumulation would proceed for larger evidence), and no speed-accuracy trade-off manipulation to test for some kind of bounded accumulation, either of which would provide stronger support that the error detection process reflects a similar sequential sampling of evidence. Thus the authors should discuss and/or address this limitation.*

We of course agree that an experimental manipulation of error evidence strength would provide additional support for our conclusion that the CPP reflects the process of post-decisional accumulation. However, it is unclear how such a manipulation might be successfully implemented. One option could be to vary the duration following the first-order response that the imperative stimulus is presented for, but we question the extent to which post-response processing in such situations reflects a truly second-order metacognitive process. By contrast, a central aspect of our results is the demonstration that the decision signals that we describe persist for an extended period of time after first-order response execution even in the explicit absence of any new sensory input. This approach allowed us to verify that truly metacognitive error detection decisions are preceded by a CPP that exhibits the critical properties of a build-to-threshold decision variable as specified by sequential sampling models (an RT-predictive build-up rate and threshold amplitude at response), which itself constitutes the strongest neurophysiological support to date for the existence of post-decision accumulation. Moreover, in doing so we were able to show that this process is highly sensitive to variation in the strength of internal conflict signals (i.e. FCθ).

Although we did not manipulate the criterion that subjects imposed on their second-order decisions (e.g. via manipulation of speed-accuracy instructions or of the payoff matrix for second-order decisions), we note that this approach has been taken in a previous investigation of the Error Positivity (Pe) component (Steinhauser & Yeung, 2010) and the resulting effects were entirely consistent with our proposal that this signal reflects a second-order accumulation process (specifically, a higher criterion was associated with larger trial-averaged Pe signals). As we outline in the Discussion section (paragraph seven), however, that study could not identify this signal with the mechanism of evidence accumulation, primarily because it enforced a delay on second-order responding. This point highlights what we believe to be another important strength of our manuscript: its usefulness for clarifying the mechanistic significance of previous reports on the Pe.

We thank the reviewer for these comments and we have integrated the points above into the revised manuscript (Discussion).

*3) Some analytic choices appear to be unrelated to other similar analyses, raising worrying concerns of tailoring the final report to the best outcomes of multiple choices rather than relying on the output of a single analytic framework. This decreases confidence in the robustness and thus the replicability of these full sets of findings and raises concerns of "p-hacking" based on the best of many seemingly arbitrary analytic choices (which I do not think are arbitrary at all). In particular, in paragraph four of the subsection “EEG acquisition and preprocessing” the authors describe the use of differing baseline procedures for frontal theta, which is potentially problematic. As the authors allude to ("leaving within-condition trial-by-trial fluctuations in baseline power intact"), condition-specific pre-stimulus baselines could have important influence on the appropriate level of trial-to-trial vs trial-specific inference, yet it could also simply add spurious variance into the dB ratio of post-event power. The authors do not distinguish between these possibilities or otherwise adequately justify their approach. Moreover, the between-subjects contrasts were not baseline corrected and the authors describe how both are "important source(s) of variance contributing to the reported effects". The authors need to do a more compelling job of justifying their analytic choices based on a principled, a priori framework, for example a common cross-condition baseline for all contrasts (see Cohen 2014: Analyzing Neural Time Series Data, MIT Press).*

We thank the reviewers for communicating these valid concerns and now report several additional analyses that we hope will allay them. In two new figure supplements (Figure 3—figure supplement 2; Figure 6—figure supplement 3), we show the following:

1) The effect of error detection on trial-averaged FCθ power was almost identical when no baseline was applied to the theta waveforms (compare Figure 3 to Figure 3—figure supplement 2), indicating that this effect did not depend on what baseline was employed.

2) Although the within-subjects relationship between FCθ and RTd was clearly still present when a single-trial baseline was employed (Figure 3—figure supplement 2), the effect was weaker than that observed when a trial-averaged baseline (or equivalently, no baseline) was used (Figure 3—figure supplement 2). This motivated our statement in the original submission (paragraph four of the subsection “EEG acquisition and preprocessing”) that within-subjects variation in baseline power was an important source of variance contributing to the reported effects. We do not show it here but single-trial regressions of RTd on baseline theta power alone yielded a significant effect of baseline power at the group level (p=0.014).

3) The between-subjects correlation between FCθ power and drift ratio was no longer statistically significant (p=0.16) when the FCθ signals were baselined for each subject (Figure 6—figure supplement 3). This motivated our statement in the original submission that between-subjects variation in baseline power appeared to be an important source of variance contributing to the observed between-subjects FCθ power effect.

We note that, in light of the above, our baselining protocol is principled insofar as it preserves the relevant sources of variation in baseline power for each analysis. We hope that this convinces the reviewers that we were not engaged in p-hacking.

To mitigate any remaining concerns regarding the between-subjects FCθ effect, we point out that the additional between-subjects noise that is present when not employing a baseline should in principle work to make any existing effects harder to detect – it seems less likely, then, that the lack of a baseline for this analysis is actually causing a spurious effect, and more reasonable (and consistent with at least a subset of the within-subjects analyses; Figure 3—figure supplement 2) that variation in baseline power is actually meaningful. Moreover, we also now report (paragraph three of the subsection “Drift diffusion modelling of the second-order decision process”) using partial correlation that the FCθ/drift ratio correlation is still present (r=0.52, p=0.007) when unbaselined θ power averaged over bilateral posterior electrodes (P7/P8) was included as a co-variate (Cohen & Donner, 2013). This result suggests that the reported correlation is not driven by spurious ‘global’ differences in oscillatory power across subjects but is specific to the fronto-central signal of interest here.

*4) In the subsection “Signal interactions predict the timing of error detection” and in paragraph four of subsection “Analysis of electrophysiological decision signals”: The authors need to specify their regression models in greater detail. They report that "single-trial regressions revealed that this effect was only marginally significant after accounting for the contribution of CPP build up rate". However, it isn't made explicitly clear if they are included in a sequential or type III sum of squares manner to account for variance in main effects prior to the other main effects.*

We apologize for the lack of clarity here. We took the type III sum of squares approach – in a given model, all regressors were entered simultaneously and the associated regression coefficients represent the unique portion of variance in the dependent variable accounted for by each regressor, independent of all other regressors. We have now made this explicit in our Methods description of the single-trial regressions (paragraph four of the subsection “Analysis of electrophysiological decision signals”). We also slightly changed the phrasing in the Results section highlighted by the reviewers: rather than “[…]after accounting for the contribution[…]” (which may imply a sequential approach), we rephrase to “[…]when the contribution of CPP build-up rate was accounted for.”

*5) Paragraph one of subsection “Analysis of electrophysiological decision signals”: One-tailed test are rarely appropriate for hypotheses that could possibly go in either direction. Here, it is entirely possible that there would be a linear decline in an EEG signal associated with longer RTs.*

We had strong a priori motivations for using one-tailed tests in this instance: we focus on two signals (CPP, FCθ) that are invariably observed to increase rather than decrease in magnitude prior to response, and we predicted based on our previous research (Kelly & O’Connell, 2013; Twomey et al., 2015) and the logic of accumulation-to-bound sequential sampling models that if one/both of these signals reflects the process of evidence accumulation, their build-up rate should specifically be steeper for faster RTs. We also note that these analyses are primarily illustrative, and we do also provide a more rigorous test of the build-up rate/RT relationships with our two-tailed single-trial regression analysis (Figure 4). These points notwithstanding, we have re-plotted all CPP and FCθ waveforms binned by primary RT and RT_d_ below (Figure 8), this time using two-tailed tests for signal build-up rates. As can be seen, this adjustment makes no substantive difference to the presence of all of the important build-up rate effects that we highlight in the manuscript.

Author response image 1.Re-plotting of panels from Figure 2, Figure 2 and Figure 3 from the main manuscript, using two-tailed rather than one-tailed tests on signal build-up rates (significance indicated by black running markers).**DOI:**
http://dx.doi.org/10.7554/eLife.11946.021

*6) The inference of a causal link between FCθ and CPP seems on shaky ground. Comparing Figure 2 and Figure 3, FCθ power differentiates between RTd earlier than CPP, but could this be due to the influence of neighbouring timepoints on the wavelet measure? This could be addressed with narrow-band wavelets or a filter-HIlbert approach. In single-trial analyses the FCθ power window (-0.1 to +0.4s) extends beyond the window at which the CPP build-up rate is calculated (+0.1 to +0.3s; Methods, paragraph three of the subsection “Analysis of electrophysiological decision signals”). It's therefore odd to enter FCθ as a cause of CPP in a mediation analysis given that the relevant part of the former potentially takes place after the latter. Putting aside the issue of timing, could the assumed causality be further bolstered by flipping the nodes in the mediation analysis, i.e. could the authors show that FCθ does not mediate the influence of CPP build-up on RTd?*

In fact, all FCθ results with the exception of Figure 3 (time-frequency plot) were indeed obtained using a filter-Hilbert approach to bolster the temporal specificity of the analyses. We apologize for not making this clearer in the manuscript and have now done so in the legend for Figure 3. Note that we also reported in the original manuscript that results from the wavelet convolution and filter-Hilbert methods were qualitatively very similar (paragraph four of the subsection “EEG acquisition and preprocessing”).

The reviewers raise a good point about the time-windows employed for the measurement of FCθ power and CPP build-up rate in our original mediation analysis. To address this point, we carried out a new mediation analysis using the same measurement window for FCθ, but now a detection-locked window for CPP build-up rate (from -300 to -100ms relative to the error detection report on each trial). Thus, the new CPP build-up rate window was temporally dissociated from the FCθ power window across trials and, in the majority of cases, occurred later in time. The results from this mediation analysis, which we now report in the manuscript (in the subsection “Signal interactions predict the timing of error detection”), were essentially the same as our original attempt: all path coefficients were statistically significant at p<0.001, as was the critical mediation effect (see new Figure 5).

*7) The first-order CPP increases for both go and no-go trials, consistent with the notion of a domain-general accumulator. But the second-order CPP increases only on error trials (Figure 2). If the second-order decision is analogous to the first (i.e. a race between neural subpopulations controlling whether or not to press the button based on fluctuating evidence), then presumably there should also be accumulation towards a no-go bound in the second-order, confident-in-being-correct case. Doesn't this mismatch between first-order and second-order CPP features weaken the conclusion that the same DV mechanism underpins first- and second-order decisions? It would be interesting to hear the author's perspective on this. For instance the second-order CPP could reflect evidence accumulation from endogenous activities towards an overt error detection response, but the authors also argue the first-order CPP is also involved in evidence accumulation leading to a non-response (Figure 2—figure supplement 1).*

The reviewers again highlight some very pertinent issues that warrant further exposure in the manuscript. In fact, while we do assert that the same neural signal traces first- and second-order decision processes, we do not contend that the two processes are equivalent in every respect. For their first-order task, we asked participants to make discrimination decisions about the state of a stimulus: should a prepotent task response be either executed or inhibited, given the current stimulus information. These decisions required active processing of the stimulus information presented on each trial and, as we (subsection “Task behaviour”) and the reviewers point out, are typically understood in terms of a race between two competing accumulators – one for ‘Go’ and one for ‘Don’t Go’ (Boucher et al., 2007; Gomez et al., 2007; Logan & Cowan, 1984; Verbruggen & Logan, 2009). In this kind of context, where two distinct alternatives must be arbitrated between given the current sensory information, we have previously shown that the CPP reflects the evidence accumulation process for decisions in favor of either alternative (Kelly & O’Connell, 2013). By contrast, for their second-order task of error detection, participants needed to engage in a process that is akin to signal detection: if there is sufficient evidence available that an error has been committed, then simply execute a response to indicate this. We (O’Connell et al., 2012) and others (e.g. Hillyard et al., 1971) have shown that the CPP/P3 in such detection contexts is clearly evident for hits (detected errors in the present context) but absent for misses (undetected errors).

Moreover, there is strong evidence to suggest that discrimination and detection decisions are processed differently in the brain, in ways that corroborate our above distinction between the first- (discrimination) and second-order (detection) decision processes on our paradigm. Much neurophysiological data indicates that two- or multi-alternative discriminations invoke a process of neural evidence gathering that manifests in build-to-threshold dynamics in competitive populations of neurons that are selective for each choice alternative (e.g. Gold & Shadlen, 2007) – that is, any given decision will be preceded by an evoked increase in activation of one pool of neurons. By contrast, simple perceptual detection tasks are thought to be determined by a competition between ‘No/stimulus absent’ and ‘Yes/stimulus present’ pools of neurons (e.g. Deco et al., 2007). The ‘No’ pool receives a static input and dominates the competition by default, whereas the ‘Yes’ pool is activated by stimulus information (or in our scenario, by ‘error evidence’). On hit trials (detected errors), the 'Yes' pool eventually dominates the competition and becomes highly activated; on miss trials (undetected errors), it fails to do so and there ends up being little-to-no evoked increase in firing rate in either pool. Thus, discrimination and detection decisions are clearly distinguishable in terms of neural dynamics.

We believe that the dissociable dynamics of the first-order (domain-general) and second-order (selective to error evidence) CPPs that we observed serve to highlight a critical difference in the nature of the two decision processes. The above considerations motivated our statements in the original manuscript that the second-order decision process appeared to be qualitatively distinct from the first-order process and involved the selective accumulation of error evidence (paragraph five of the subsection “A centro-parietal signature of first- and second-order evidence accumulation”). However, we now recognize that these issues, and in particular the distinction between discrimination and detection decisions, warrants further unpacking and we have now attempted to elaborate appropriately in the Discussion section of the revised submission (paragraph three).

[Editors' note: further revisions were requested prior to acceptance, as described below.]

*Reviewer #2:*

*The authors have mostly addressed my previous concerns in the first round of reviews. The one part that I don't think has been fully addressed is the question of whether there are actually differences between undetected errors and go trials in the data that would not be captured by the model (in general I'm happy with the response to point 1 that the connection between first- and second-order accumulation can be left for future work, but it seems important to fully display the data features that might inform such work). For instance the ERP analysis in Figure 3—figure supplement 1 only directly contrasts undetected and detected errors, but there appears to be a difference between go and undetected errors later in the time course. Similarly only go-trial build-up rates are documented in Figure 2 and Figure 3. I suggest including the comparison between go and undetected CPP and FCθ timecourses as a supplemental figure to cover this issue.*

We apologize for this omission and thank the reviewer for again bringing it to our attention. We have now included an additional figure supplement as recommended by the reviewer (labelled ‘Figure 3—figure supplement 1’ in the latest submission), which displays direct comparisons of the CPP, fronto-central ERP and FCθ signals on Go trials and undetected errors. These analyses did not yield clear and robust effects for the CPP and FCθ signals (panels A and C of the new supplement), but as the reviewer correctly points out, there is a clear difference in fronto-central ERP amplitude that manifests from approximately 400ms post-response (panel B). We do not have a clear hypothesis about the functional significance of this effect and do not discuss it in detail in the revised submission. But we do acknowledge that the effect hints that these trial types may be at least partially dissociable in terms of underlying neural dynamics. We now draw attention to this interesting point in the revised manuscript (Discussion, paragraph six) in the context of our discussion about the inability of our model to provide a precise explanation for the apparent equivalence of the CPP on Go trials and undetected errors.